# Task state representations in vmPFC mediate relevant and irrelevant value signals and their behavioral influence

Nir Moneta [1,2,3] ✉, Mona M. Garvert[1,2,4], Hauke R. Heekeren[3,5,6] & Nicolas W. Schuck [1,2,6] ✉

The ventromedial prefrontal-cortex (vmPFC) is known to contain expected value signals that inform our choices. But expected values even for the same stimulus can differ by task. In this study, we asked how the brain flexibly switches between such value representations in a task-dependent manner. Thirty-five participants alternated between tasks in which either stimulus color or motion predicted rewards. We show that multivariate vmPFC signals contain a rich representation that includes the current task state or context (motion/color), the associated expected value, and crucially, the irrelevant value of the alternative context. We also find that irrelevant value representations in vmPFC compete with relevant value signals, interact with task-state representations and relate to behavioral signs of value competition. Our results shed light on vmPFC's role in decision making, bridging between its role in mapping observations onto the task states of a mental map, and computing expected values for multiple states.

Decisions are always made within the context of a given task. Even a simple choice between two apples will depend on whether the task is to find a snack, or to buy ingredients for a cake. In other words: the same objects can yield different outcomes in different task contexts. This could complicate the computations underlying retrieval of learned values during a decision, since outcome expectations from the wrong context might exert influence on the neural representation of the available options.

Which reward a choice will yield in a given task context is at the core of many decisions (e.g. ref. 1). Ventromedial prefrontal cortex (vmPFC) represents this so-called expected value (EV) in a variety of species[2–7], and thereby is crucial in determining choices[8]. Several investigations have also shed light on how the brain maps from complex sensory input to expected values, and the associated cognitive control processes. It is known, for instance, that the brain's attentional control network enhances the processing of features that are relevant

given the current task context or goal[9,10], which in turn helps shape which features influence expected value representations in vmPFC[11–16]. Moreover, vmPFC seems to also represent expected value of different features in a common currency[17,18]; and is involved in integrating reward expectations from different features of the same object[19–22]. It remains unclear, however, how context-irrelevant value expectations of available features, i.e., rewards that would be obtained in a different task-context, might affect vmPFC signals, and how such "undue" influence relates to wrong choices.

This is particularly relevant because we often have to do more than one task within the same environment, such as shopping in the same supermarket for different purposes. Cognitive control processes are known to arbitrate between relevant and irrelevant information[23,24], and it has been suggested that they also gate the flow of information within the value network[22,25]. But although cognitive control does gate relevant information, it is also known that task-switching leads to less

[1]Max Planck Research Group NeuroCode, Max Planck Institute for Human Development, 14195 Berlin, Germany. [2]Max Planck UCL Centre for Computational Psychiatry and Ageing Research, Berlin, 14195 Berlin, Germany. [3]Einstein Center for Neurosciences Berlin, Charité Universitätsmedizin Berlin, 10117 Berlin, Germany. [4]Department of Psychology, Max Planck Institute for Human Cognitive and Brain Sciences, 04103 Leipzig, Germany. [5]Department of Education and Psychology, Freie Universität Berlin, 14195 Berlin, Germany. [6]Institute of Psychology, Universität Hamburg, 20146 Hamburg, Germany. ✉e-mail: moneta@mpib-berlin.mpg.de; schuck@mpib-berlin.mpg.de

than perfect separation between task contexts/goals[24] and results in processing of task-irrelevant aspects[23]. Several studies found traces of the distracting features in several cortical regions, including areas responsible for task execution[26–30]. Similarly, not only task-relevant but also task-irrelevant valuation has been shown to influence cognitive control[31,32] as well as activity in vmPFC[33] and posterior parietal cortex[34]. We therefore hypothesized that during choice the vmPFC will represent different values that occur in different task contexts, i.e., values appropriate in the current context, as well as other, context-inappropriate and therefore choice-irrelevant values. Importantly, unlike in standard cognitive control settings, we asked whether the above-mentioned control during value-based choice involves the arbitration between the expected values that would result from the counterfactual choices one would have made in another context.

If that is the case, the neural representation of context might play a major role in gating context-dependent values in vmPFC. Previous work has shown that vmPFC is involved in representing such context-signals[35–38], which suggests that its role goes beyond representing attention-filtered values. Note that knowing the current context alone will not immediately resolve which value of two presented options should be represented, similar to how knowing what you are shopping for (cake or snack) will not answer which of the available apples you should pick. We therefore hypothesized that vmPFC would have a role that goes beyond only encoding the task context, namely that it would also be involved in the arbitration between context-dependent values, meaning that a stronger activation of the relevant task-context will also enhance the representation of task-relevant values. Such a multi-faceted representation of multiple values and task contexts within the same region would reconcile work that emphasizes the role of choice value representations in vmPFC and orbitofrontal cortex (OFC)[2–8] with work which emphasizes the encoding of other aspects of the current task[39–43], in particular of so-called task states[35–38], within the same region (see also refs. [44,45]). More specifically, we propose that context/task state representations influence value computations in vmPFC, such that a state representation triggers a comparison between the values of options as they would be expected in the represented state/context. In consequence, the value of the option that would be best in the activated state will become represented, and partial co-activation of different possible states could therefore lead to value representations that can refer to different choices (the value of the apple best for snacking and the value of the apple best for baking, even if those are different apples). An alternative view in which state representations do not impact value computations would assume that activated values would always refer to the choice one is going to make in the present context (how valuable the apple chosen for snaking would be for baking).

We investigated these questions using a multi-feature choice task in which different features of the same stimulus predicted different outcomes, and a task-context cue modulated which feature was relevant. We show that participants compute both value expectations of the relevant context as well as value expectations of an additional, explicitly cued-to-ignore, irrelevant context. Behavioral analyses indicated a choice conflict modulated by the possible expected values of the relevant and irrelevant context. Multivariate fMRI signals in a vmPFC value ROI were sensitive to (1) relevant values, (2) contextually irrelevant values and (3) the identity of the current context. We also found that increased representation of irrelevant values during choice were accompanied by a decreased representation of the relevant values, indicating a value competition in vmPFC. This competition was modulated by the task-context signal found in vmPFC. Lastly, we found that neural indicators of context, values and the competition between them were linked to increased choice conflict. We suggest that information within the vmPFC is organized into a complex multi-faceted representation in which multiple values of the same choice under different task-contexts are co-represented and compete in guiding

behavior, while a context (or state) signal might act as a moderator of this competition.

## Results

### Behavioral results

Thirty-five right-handed young adults (18 women, $\mu_{age} = 27.6$, $\sigma_{age} = 3.35$, see "Methods" for exclusions) were asked to judge either the color (context 1) or motion direction (context 2) of moving dots on a screen (random dot motion kinematograms (e.g. ref. [46])). Four different colors and motion directions were used. Before entering the MRI scanner, participants performed a stair-casing task in which participants first received a cue that instructed them which feature (a color or direction) will be the target of the current trial. Then participants had to select the matching stimulus from two random dot motion stimuli (see Fig. S1c). In this task, motion-coherence and the speed which dots changed from gray to a target color were adjusted such that the different stimulus features could be discriminated equally fast, both within and between contexts (i.e., Color/Motion, Fig. S1c). As intended, this led to significantly reduced differences in reaction times (RTs) between the eight stimulus features, within and between contexts (paired $t$-test on RT variance before and after the staircasing: $t_{(34)} = 7.29$, $p < 0.001$, Fig. 1a), also when tested for each button separately ($t_{(34)} =$ Left: 6.52, Right: 7.70, $p$s $< 0.001$, Fig. S1d).

Only then, participants learned to associate each color and motion feature with a fixed number of points (10, 30, 50 or 70 points), whereby one motion direction and one color each led to the same reward (counterbalanced across participants, Fig. 1b). To this end, participants made choices between clouds that had only one feature-type, while the other feature type was absent or ambiguous (clouds were gray in motion-only clouds and moved randomly in color clouds). To encourage mapping of all features on a unitary value scale, choices in this part (and only here) also had to be made between contexts (e.g., between a green and a horizontal-moving cloud). Participants achieved near-ceiling accuracy in choosing the cloud with the highest valued feature ($\mu = 0.89$, $\sigma = 0.06$, $t$-test against chance: $t_{(34)} = 41.8$, $p < 0.001$, Fig. 1c), also when tested separately for color, motion and across context ($\mu = 0.88, 0.87, 0.83$, $\sigma = 0.09, 0.1, 0.1$, $t$-tests against chance: $t_{(34)} = 23.9, 20.4, 19.9$, $p$s $< 0.001$, respectively, Fig. S1e). Once inside the MRI scanner, one additional training block ensured changes in presentation mode did not induce feature-specific RT changes (Anova on mean RT for each feature: $F_{(7,202)} = 1.06$, $p = 0.392$). These procedures made sure that participants began the main task with firm knowledge of feature values; and that RT differences would not reflect perceptual differences, but could be attributed to the associated values. Additional information about the pre-scanning phase can be found in "Methods" and in Fig. S1.

During the main task, participants had to select one of two dot-motion clouds. In each trial, participants were first cued whether a decision should be made based on color or motion features, and then had to choose the cloud that would lead to the largest number of points. Following their choice, participants received the points corresponding to the value associated with the chosen cloud's relevant feature. To reduce complexity, the two features of the cued task-context always had a value difference of 20, i.e., the choices on the cued context were only between values of 10 vs. 30, 30 vs. 50 or 50 vs. 70. One third of the trials consisted of a choice between single-feature clouds of the same context (henceforth: 1D trials, Fig. 1d, top). All other trials were dual-feature trials, i.e., each cloud had a color and a motion direction at the same time (henceforth: 2D trials, Fig. 1d bottom), but only the context indicated by the cue mattered. Thus, while 2D trials involved four features in total (two clouds with two features each), only the two color or two motion features were relevant for determining the outcome. The cued context stayed the same for four to seven trials. Importantly, for each comparison of relevant features, we varied the values of the irrelevant context, such that each relevant value was

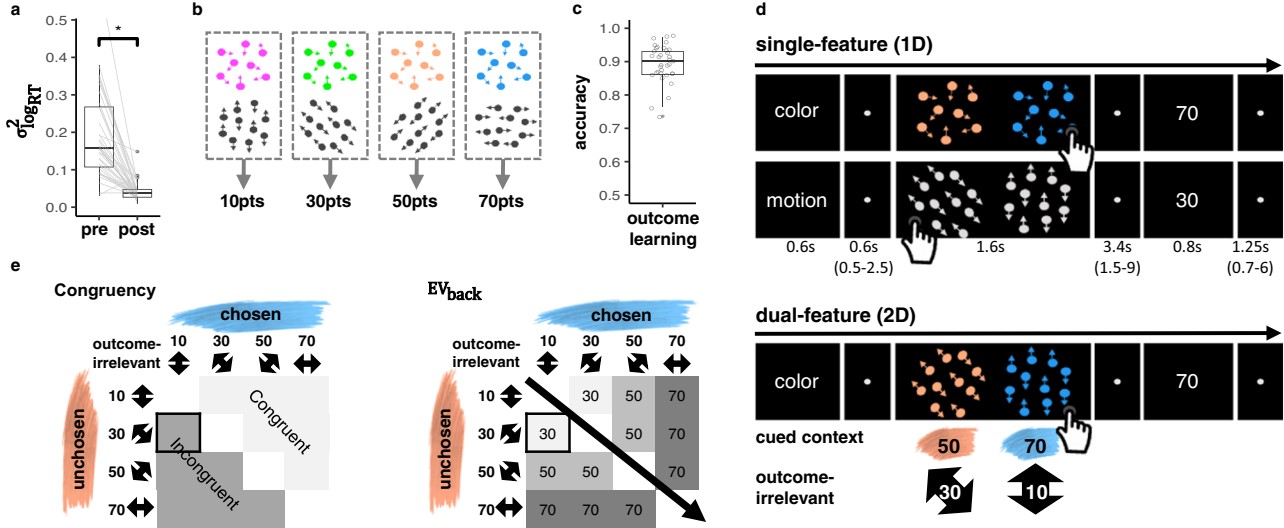

**Fig. 1 | Task and design. a** Prior to value-learning, a participant-specific staircasing procedure adjusted color and motion parameters such that variance of reaction times across different color and motion features (y-axis) was reduced (paired t-test, $p < 0.001$, $n = 35$). Box covers interquartile range (IQR), mid-line reflects mean, whiskers the range of the data (until ±1.5*IQR), and solid points represent outliers beyond whiskers. **b** After staircasing, specific rewards were assigned to each of the four color and four motion directions, such that one feature from each context was associated with the same reward/value. Feature-value mapping was counter-balanced across participants. **c** Participants achieved near ceiling accuracy in choosing the highest valued feature after training ($\mu = 0.89$, $\sigma = 0.06$, $n = 35$). Box-plot as in (**a**). **d** Single-feature (1D, top) and dual-feature (2D, bottom) trials both started with a cue of the relevant context ("Color" or "Motion", 0.6 s), followed by a fixation (0.5–2.5 s, $\mu = 0.6$ s) and a choice between two clouds (1.6 s). In 1D trials, each cloud only had one relevant feature (colored dots, but random motion, or directed motion, but gray dots), while in 2D trials each cloud had a motion and a color feature. Participants were explicitly asked to select the option yielding the highest outcome in the cued context and ignore irrelevant features. Then followed another fixation (1.5–9 s, $\mu = 3.4$ s) and the value associated with the chosen cloud's feature of the cued context (outcome, 0.8 s). The next trial started after another fixation (0.7–6 s, $\mu = 1.25$ s). **e** Experimental manipulation of irrelevant values in 2D trials. For each relevant feature pair (e.g., blue and orange), all possible context-irrelevant feature-combinations were included in the task, except same feature on both sides. Congruency (left): trials were termed congruent when irrelevant features favored the same choice as the relevant features, otherwise incongruent. EV_back (right): trials were also characterized by the hypothetical expected value of contextually-irrelevant features, i.e., the maximum value of both irrelevant features. NB that both aspects did not have any impact on outcomes and were irrelevant for the task at hand and that EV, EV_back and Congruency were orthogonal by design. Highlighted cell reflects example trial in (**d**), bottom. Source data are provided as a Source Data file.

paired with all possible irrelevant values (Fig. 1e). While the irrelevant context in a trial did not impact the outcome, it might nevertheless influence behavior. Specifically, the hypothetical outcomes as they would occur in the irrelevant context could favor the same side as the relevant one, or not (Congruent vs. Incongruent trials, see Fig. 1e left), and have larger or smaller values compared to the relevant features (Fig. 1e right).

We investigated the impact of these factors on RTs in correct 2D trials, where the extensive training ensured near-ceiling performance throughout the main task ($\mu = 0.91$, $\sigma = 0.05$, t-test against chance: $t_{(34)} = 48.48$, $p < 0.0001$, Fig. 2a). RTs were log transformed to approximate normality and analyzed using mixed effects models with nuisance regressors for choice side (left/right), time on task (trial number), differences between attentional contexts (color/motion) and number of trials since the last context switch (all nuisance regressors had a significant effect on RTs, Type II Wald $\chi^2$ test, all $p$s < 0.03). We used hierarchical model comparison to assess the effects of (1) the objective value of the chosen option (or: EV), i.e., points associated with the features on the cued context; (2) the maximum points that could have been obtained if the irrelevant features were the relevant ones (the expected value of the background, henceforth: EV_back, Fig. 1e right), and (3) whether the irrelevant features favored the same side as the relevant ones or not (Congruency, Fig. 1e left). Any effect of the latter two factors would indicate that outcome associations that were irrelevant in the current context nevertheless influence behavior, and therefore could be represented in vmPFC.

We found that participants reacted faster in trials that yielded larger rewards and slower in incongruent compared to congruent trials (likelihood-ratio test to asses improved model fit, EV: $\chi^2_{(1)} = 1374.6$,

$p < 0.001$, Congruency: $\chi^2_{(1)} = 29.0$, $p < 0.001$, Fig. 2b, c). Moreover, compared to 1D trials, participants were slower to respond to incongruent trials and faster to respond to congruent trials (paired t-tests: $t_{(34)} = -2.79$, $p = 0.013$, $t_{(34)} = 2.5$, $p = 0.017$ respectively, FDR-corrected, see Fig. 2b, c). Crucially, we found that Congruency interacted with the expected value of the other context: larger EV_back increased participants' speed on congruent trials and had the opposite effect on incongruent trials (LR-test: $\chi^2_{(1)} = 18.19$, $p < 0.001$, Fig. 2d). These effects show that even when participants chose accurately based on the relevant context, the information of the irrelevant context was not completely filtered. The expected value of a "counterfactual" choice resulting from consideration of the irrelevant context mattered: the outcome such a choice could have led to influenced reaction times. A full model description including effect sizes and confidence intervals can be found in SI Table S2.

Neither adding a main effect for EV_back nor the interaction of EV × EV_back improved model fit (LR-tests: $\chi^2_{(1)} = 1.21$, $p = 0.27$, $\chi^2_{(1)} = 0.01$, $p = 0.9$ respectively), indicating that neither the presence of larger irrelevant values alone, nor their similarity to the relevant values influenced participants' RTs. Additionally, the lower valued irrelevant feature did not show comparable effects and did not interact with Congruency (LR-test to baseline model: $\chi^2_{(1)} = 0.92$, $p = 0.336$, with interaction: $\chi^2_{(1)} = 2.76$, $p = 0.251$). Replacing EV_back with a parameter of overall value of the irrelevant features did not improve the fit (which could be understood as an overall distraction of the irrelevant context, AIC of model with EV_back × Congruency: −6626.649, AIC of model with Overall value × Congruency: −6619.878, Fig. S3). These results further support that it is specifically the expected reward of the ignored context that played a role in participants' RT.

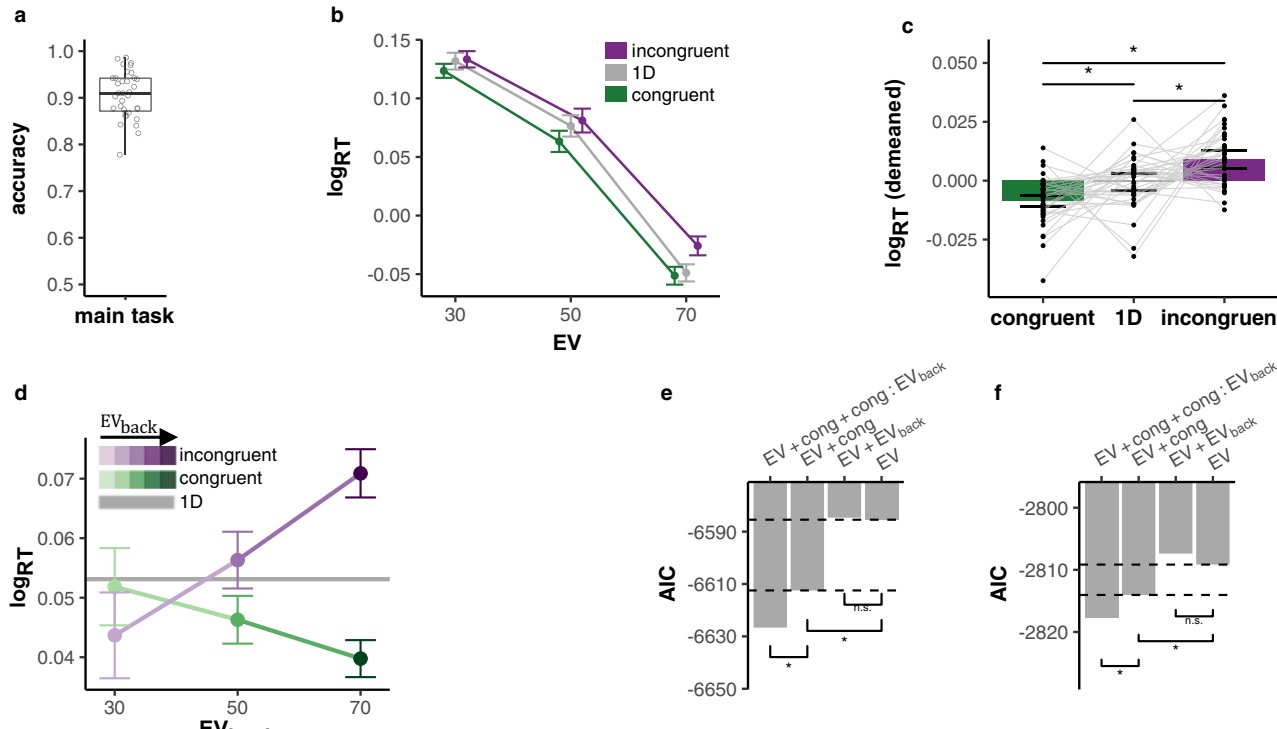

**Fig. 2 | Behavioral results. a** Participants performed near-ceiling throughout the main task, $\mu = 0.905$, $\sigma = 0.05$ ($n = 35$). Box covers interquartile range (IQR), mid-line reflects mean, whiskers the range of the data (until ±1.5*IQR), and solid points represent outliers beyond whiskers. **b** Participants reacted faster to higher Expected Values (EV, $x$-axis) and slower to incongruent (purple) compared to congruent (green) trials. RTs for 1D trials shown in gray. Error bars represent corrected within subject SEMs[102,103]. **c** Comparison of log RTs by trial condition. Incongruent trials were slower than 1D trials (paired $t$-test: $p = 0.013$), and 1D trials slower than congruent trials (paired $t$-test: $p = 0.017$; paired $t$-test congruent vs. incongruent: $p < 0.001$). Error bars represent corrected within subject SEMs[102,103]. $p$ values FDR-corrected, $n = 35$. **d** The Congruency effect was modulated by $EV_{back}$, i.e., the more participants could expect to receive from the ignored context, the slower they were when the contexts disagreed and respectively faster when contexts agreed ($x$-axis,

shades of colors). Likelihood-ratio test (LRR) to asses improved model fit: $p < 0.001$, $n = 35$. Gray horizontal line depicts the average RT for 1D trials across subjects and EV. Error bars as above. **e** Hierarchical comparison of 2D trial log-RT models showed that inclusion of a Congruency main effect ($p < 0.001$, see **c**), yet not $EV_{back}$ ($p = 0.27$), improved model fit. However, including an additional Congruency × $EV_{back}$ interaction improved model fit even more ($p < 0.001$, see **d**). $p$ values from LR tests as above, stars indicate $p < 0.05$, $n = 35$. **f** We replicated the behavioral results in an independent sample of 21 participants outside the MRI scanner. Including Congruency ($p = 0.009$) but not $EV_{back}$ ($p = 0.63$), improved model fit. Including an additional Congruency × $EV_{back}$ interaction explained the data best ($p = 0.017$). $p$ values/stars as in (**e**). Source data are provided as a Source Data file.

All major RT effects hold when running the models nested within levels of EV, Block Context or switch (Fig. S2). Moreover, the number of trials since context switch did not interact with our main effect (LR-test with added term for Congruency × $EV_{back}$ × switch: $\chi^2_{(1)} = 3.70$, $p = 0.157$) and our main RT effects still hold when we excluded the first two trials after the context switch (LR-tests: Congruency, $\chi^2_{(1)} = 8.12$, $p = 0.004$, Congruency × $EV_{back}$, $\chi^2_{(1)} = 16.61$, $p < 0.001$). We note that an interaction of EV × Congruency indicated stronger Congruency effect for higher EV (LR-test with added term: $\chi^2_{(1)} = 4.34$, $p = 0.037$, Fig. 2b), but did not replicate in the replication sample (see below, $\chi^2_{(1)} = 0.23$, $p = 0.63$). Details of other significant effects and alternative models considering for instance within-cloud or between-context value differences can be found in Figs. S3 and S4, respectively.

We replicated these findings in an additional sample of 21 participants (15 women, $\mu_{age} = 27.1$, $\sigma_{age} = 4.91$) that were tested outside of the MRI scanner (LR-tests: Congruency, $\chi^2_{(1)} = 6.89$, $p = 0.009$, $EV_{back}$, $\chi^2_{(1)} = 0.23$, $p = 0.63$, Congruency × $EV_{back}$, $\chi^2_{(1)} = 5.69$, $p = 0.017$, Fig. 2e).

We next modeled choice accuracy in 2D trials using the same analysis approach and nuisance variables (see "Methods" and Fig. S5) and found the same effects as the RT models: (1) Higher accuracy for higher EV (LR-test: $\chi^2_{(1)} = 14.61$, $p < 0.001$) (2) decreased performance on incongruent trials with (3) higher error rates occurring on trials with higher $EV_{back}$ (LR-tests: $\chi^2_{(1)} = 66.12$, $p < 0.001$, $\chi^2_{(1)} = 6.99$, $p = 0.03$, respectively, Fig. S5).

In summary, these results indicated that participants did not merely perform a value-based choice among features on the currently relevant context. Rather, both reaction times and accuracy indicated that participants also retrieved the values of irrelevant features and computed the resulting counterfactual choice. We next turned to test if the neural code of vmPFC would also incorporate such counterfactual choices, and if so, how the representation of the relevant and irrelevant contexts and their associated values might interact.

## fMRI results

**Outcome-relevant and outcome-irrelevant values co-exist within the vmPFC.** We derived a value-sensitive vmPFC ROI following common procedures in the literature (e.g. refs. 4,5) (see Fig. 3a and "Methods") and tested whether both relevant and irrelevant expected values are reflected in multivariate vmPFC patterns using RSA. To estimate value-related activity patterns within the vmPFC mask, we fitted a general linear model (GLM) with one separate regressor for each combination of EV and $EV_{back}$, irrespective of the context (cross-validated, 1D trials modeled separately). After multivariate noise normalization and mean pattern subtraction (see ref. 47) we computed the Mahalanobis distance between each combination of regressor. This resulted in one 9 × 9 representational dissimilarity matrix (RDM, Fig. 3 and "Methods") per subject, which we analyzed using mixed effects models (Gamma family with a inverse link[48]). We first asked

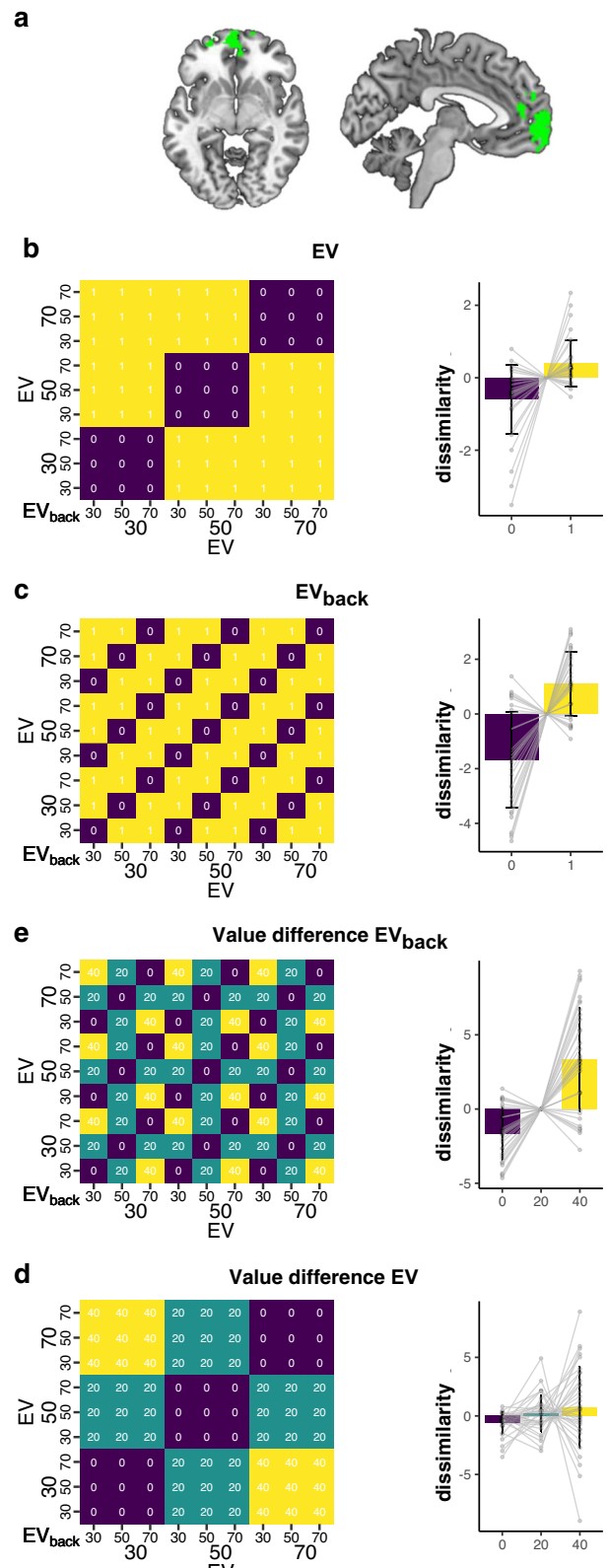

**Fig. 3 | RSA analyses show that vmPFC encodes both relevant as well as irrelevant expected values given the current task context. a** vmPFC region used in all analyses (green voxels), defined functionally as the positive effect of a univariate value regressor thresholded at $p_{FDR} < 0.0005$ (one sided $t$-test, see "Methods"). Note that no information regarding the contextually irrelevant values was used to construct the ROI. Axial slice (left) at $x = -6$; Sagittal slice (right) at $z = -6$. **b** Left: Model RDM, each cell represents one combination of EV and $EV_{back}$, see axes. Colors reflect whether a combination of trials had the same EV (purple) or not (yellow). Right: Dissimilarity of vmPFC activation patterns for trials with the same vs. different EV. Dissimilarity was lower in trials that share the same expected value (EV, $p < 0.001$, $n = 35$). **c** Model RDM (left) testing whether irrelevant expected value ($EV_{back}$) affected similarity in vmPFC. We found less dissimilarity for trails with the same $EV_{back}$ ($p < 0.001$, $n = 35$, right). **d** Left: Model RDM that tested whether patterns similarity was influenced by the size of EV differences (0: purple, 20: turquoise, 40: yellow). Right: Average dissimilarity associated with the varying levels of value difference, indicating that larger EV differences between trials were related to higher pattern dissimilarity ($p < 0.001$, $n = 35$). **e** The same effect was found with respect to $EV_{back}$ where patterns that share the same $EV_{back}$ (irrespective to EV) also showed a decrease in dissimilarity ($p < 0.001$, $n = 35$). Data shown in bar plots are demeaned by trial-frequency in the design to match the mixed effect models (see "Methods" and Fig. S6). Error bars in (**b**)–(**e**) represent corrected within-subject SEMs[102,103]. $p$ values in (**b**)–(**e**) reflect likelihood-ratio test of improved model fit, see main text. Source data are provided as a Source Data file.

when sharing $EV_{back}$ and 1 otherwise) further improved model fit (LR-test with added term: $\chi^2_{(1)} = 247.67$, $p < 0.001$, Fig. 3c).

We then reasoned that the neural codes of expected values should also reflect value-differences in a gradual manner. We therefore asked whether pattern similarity was not only increased if two trials had the same value (e.g., comparing "30" to "30", Fig. 3d purple cells), but also higher when the values in two trials had a difference of 20 (e.g., "30" to "50", Fig. 3d turquoise) compared to a value difference of 40 (e.g., "30" to "70", Fig. 3d yellow). Indeed we found that adding main effects for the value difference of EV as well as $EV_{back}$ improved model fit ($VD_{EV}$: LR-test compared to a null model: $\chi^2_{(1)} = 12.34$, $p < 0.001$, $VD_{EV_{back}}$: LR-test with added term: $\chi^2_{(1)} = 256.98$, $p < 0.001$, Fig. 3c, d). Note that the full model with both value difference effects resulted in a better (lower) AIC score than the model with both main effects of the EVs (AIC = 165,231 and AIC = 165,241, respectively, Fig. S6) indicating that the value similarity effect is not merely driven by the diagonal. Full models including effect sizes and confidence intervals can be found in SI Tables S5 and S6.

Hence, neural patterns in vmPFC were affected by contextually-relevant as well as irrelevant value expectations. Notably, the values of irrelevant features were computed despite being counterfactual (not related to the choice), and co-existed with well known expected values signals in vmPFC.

**vmPFC value and context signals co-exist and are positively related.** We next turned to investigate how the neural value representations of EV, $EV_{back}$ and context interacted with each other on a trial-wise level. We therefore trained a multivariate multinomial logistic regression classifier on the fMRI images acquired ~5 s after stimulus onset in same vmPFC ROI used above. An expected value classifier was trained on behaviorally accurate 1D trials, where no irrelevant values were present (henceforth: Value classifier, Fig. 4a, left; leave-one-run-out training; see "Methods"). For each testing example, the classifier assigned the probability of each class given the data (classes are the expected outcomes, i.e., "30", "50" and "70", and probabilities sum up to 1, Fig. 4a, right). Crucially, it had no information about the task context of each given trial (training sets were up-sampled to balance w.r.t. color/motion contexts, see "Methods"). We first validated that the classifier was sensitive to values, as expected given the nature of the ROI. Indeed, the class with the maximum probability corresponded to the objective outcome significantly more often than chance, both

whether EV was reflected in the RDMs, as expected given that we used a functionally defined value ROI. Indeed, adding a main effect for EV dissimilarity (0 when two regressors share the same EV, 1 otherwise) improved model fit compared to a null model (LR-test: $\chi^2_{(1)} = 10.89$, $p < 0.001$, Fig. 3b). Next, we asked if the activity patterns from trials with the same $EV_{back}$ were more similar than patterns reflecting different $EV_{back}$. Strikingly, adding a main effect of $EV_{back}$ dissimilarity (0

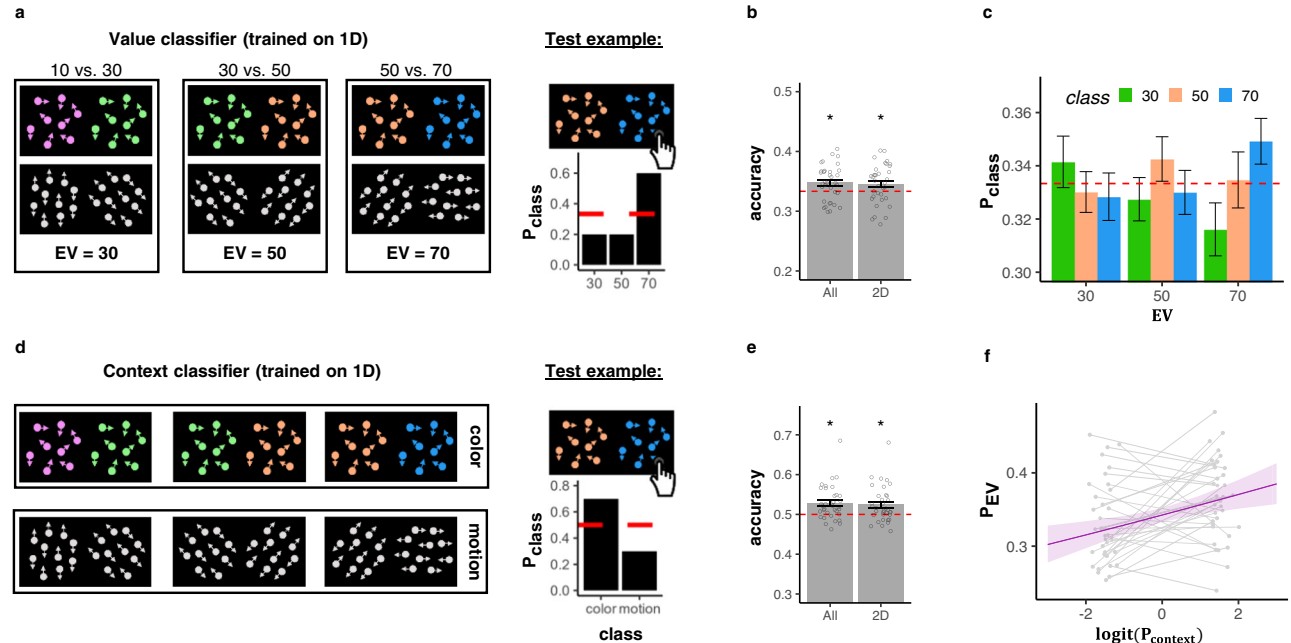

**Fig. 4 | Expected value and context signals co-reside within vmPFC. a** A logistic classifier was trained on behaviorally accurate 1D trials to predict the true EV from vmPFC patterns ("Value classifier", left). We analyzed classifier correctness and predicted probability distribution (right). shown in (**b**) and (**c**). **b** The Value classifier assigned the highest probability to the correct class (objective EV) significantly more often than chance for all trials ($p = 0.003$, $n = 35$), also when tested on generalizing to 2D trials alone ($p = 0.017$, $n = 35$). **c** The probabilities the classifier assigned to each class (y-axis, colors indicate the different classes, see legend) split by the objective EV of the trials (x-axis). As can be seen, the highest probability was assigned to the class corresponding to the objective EV of the trial (i.e., when the color label matched the X axis label). $n = 35$, for individual data points see Fig. S7. **d** A second logistic classifier was trained on the same data to distinguish between

task contexts (color vs. motion), irrespective of the EV ("Context" classifier). **e** The Context classifier assigned the highest probability to the correct class (objective Context) significantly more often than chance for all trials ($p < 0.001$, $n = 35$), also when tested on generalizing to 2D trials alone ($p = 0.001$, $n = 35$). **f** Increased evidence for the objective EV ($P_{EV}$, y-axis) was associated with stronger context signal in the same ROI (x-axis, where probabilities z-scored and logit-transformed, LR-test compared to null model: $p = 0.002$, $N = 35$). Plotted are model predictions and gray lines represent individual participants (mean of the top/bottom 20% of trials). Error bands represent the 89% confidence interval. $p$ values in (**b**) and (**e**) reflect one sided $t$-test against chance. Error bars in (**b**), (**c**) and (**e**) represent corrected within-subject SEMs[102,103]. Source data are provided as a Source Data file.

when tested on held out 1D and 2D trials as well as when tested only on 2D trials ($\mu_{all} = 0.35$, $\sigma_{all} = 0.029$, $t_{(34)} = 2.89$, $p = 0.003$, $\mu_{2D} = 0.35$, $\sigma_{2D} = 0.033$, $t_{(34)} = 2.20$, $p = 0.017$, respectively, Fig. 4b). Similar to the RSA analysis, we reasoned that the similarity between the values assigned to the classes will be reflected in gradual probability differences . Specifically, we expected not only that the probability associated with the correct class be highest (e.g., "70"), but also that the probability associated with the closest class (e.g., "50") would be higher than the probability with the least similar class (e.g., "30", Fig. 4c). Indeed we found that similar values elicited similar probabilities (LR-test of linear relation between value difference and class probability: $\chi^2_{(1)} = 12.74$, $p < 0.001$, full analysis can be found in Fig. S7). Additional control analyses indicated that our value classification results were not the result of a bias caused by overlap of perceptual features between training and test (Fig. S8).

A major feature of our task was that which value expectation was relevant depended on the task context. We therefore hypothesized that vmPFC would also encode the task context, although this is not directly value-related (the average values of both contexts were identical). We thus trained a second classifier on the same data from the EV-sensitive ROI on the same accurate 1D trials, but this time to identify if the trial was "Color" or "Motion" (Fig. 4d, left). The classifier had no information as to what was the EV of each given trial, and training sets were up-sampled to balance the EVs within each set (see "Methods"). The classifier performed above chance for decoding the correct context, again both when tested on held out trials from all conditions as well as when tested only on 2D trials ($t$-test against chance: $t_{(34)} = 3.93$,

$p < 0.001$, $t_{(34)} = 3.2$, $p = 0.001$, respectively, Fig. 4e). Moreover, the context was still decodable when keeping the perceptual input identical between the two classes (i.e., testing on 2D trials with fixed value difference of the irrelevant values of 20, since the value difference of the relevant context was always 20, $t_{(34)} = 2.73$, $p = 0.0005$).

We first hypothesized that if vmPFC is involved in signaling both context and values, then the strength of context signal might relate to the strength of the contextually relevant value. A corresponding mixed effects analysis indeed found that the probability the context classifier assigned to the correct class (henceforth: $P_{context}$) had a positive effect on the decodability of EV (henceforth: $P_{ev}$, LR-test compared to null model: $\chi^2_{(1)} = 9.12$, $p = 0.002$, Fig. 4f). In other words, the better we could decode the context, the higher was the probability assigned to the correct EV class.

In summary, we found that the Context is represented within the same region as the EV, and that the strength of its representation is directly linked to the representation of EV. The link between Context and relevant EV signals suggest that the Context signal might play a role in governing which values dominate vmPFC.

**Competition of vmPFC EV and EV_back signals is moderated by a context representation.** One main hypothesis was that contextually-irrelevant values might influence neural codes of expected value in vmPFC, and therefore should interact with EV probabilities decoded from vmPFC in a trial-wise manner. Similar to our analyses above, we used mixed effects models to test whether the Value classifier's probability of the correct class ($P_{EV}$) was influenced by EV_back and/or

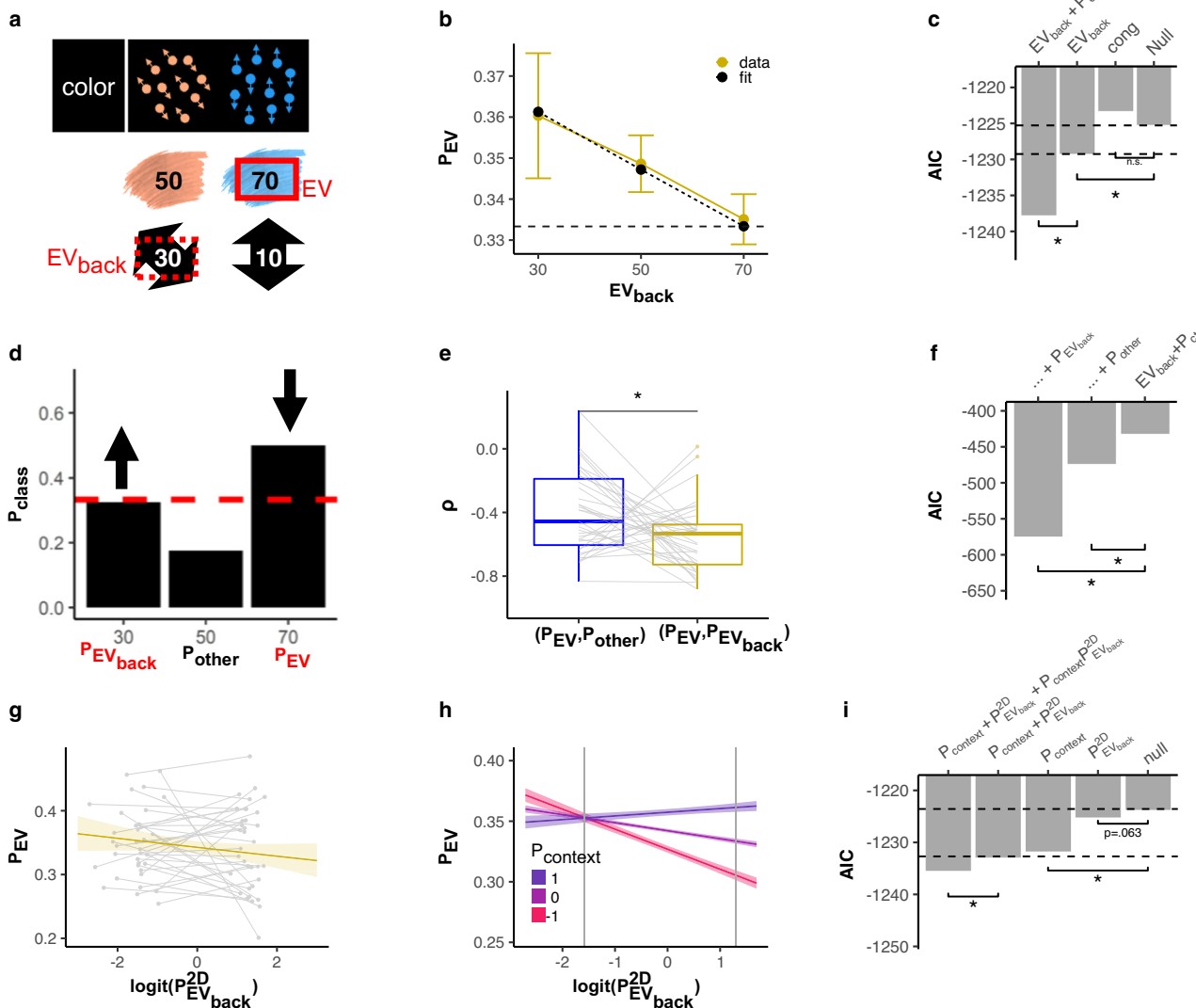

**Fig. 5 | vmPFC representations of task context, expected value and irrelevant expected value interact. a** Exemplary 2D color trial (top), its relevant outcomes (middle, color-based), and hypothetical/irrelevant outcomes (motion-based, bottom). The maxima of relevant and irrelevant outcomes are termed EV and $EV_{back}$, respectively. **b** Higher $EV_{back}$ was related to decreased decodability of EV ($P_{EV}$) in behaviorally accurate trials, likelihood-ratio (LR) test: $p = 0.015$, $n = 35$. Color see legend. Error bars represent corrected within-subject SEM. For supporting RSA evidence see Fig. S6d[102,103]. **c** Modeling the probabilities assigned to the true EV class ($P_{EV}$) showed an effect of $EV_{back}$ ($p = 0.015$) but not Congruency ($p = 0.852$). Including $EV_{back}$ and context decodability ($P_{context}$) yielded the best fit ($p = 0.001$). $p$ values reflect LR tests. **d** Illustration that Value classifier class probabilities in (**a**) example could reflect the true EV ($P_{EV}$), the $EV_{back}$ ($P_{EV_{back}}$) or neither EV or $EV_{back}$ ($P_{Other}$). **e** The correlation between $P_{EV}$ and $P_{EV_{back}}$ (yellow) was significantly more negative than the correlation between $P_{EV}$ and $P_{Other}$ (blue, paired $t$-test, $p = 0.017$, $n = 35$). Box covers interquartile range (IQR), mid-line reflects median, whiskers the range of the data (until ±1.5*IQR), and solid points represent outliers beyond whiskers. **f** Comparing models of $P_{EV}$ confirmed that adding $P_{EV_{back}}$ improved fit more than adding $P_{Other}$ (AIC: −574 vs. −473), LR test with each individual effect:

$p < 0.001$. $n = 35$. **g** The neural representations of relevant EV ($P_{EV}$, $y$-axis) and the irrelevant EV ($P_{EV_{back}}^{2D}$, $x$-axis, $z$-scored and multinomial-logit-transformed) were marginally negatively associated (LR-test: $p = 0.063$, $n = 35$). Error bands represent 89% confidence interval and gray lines individual participants' top/bottom 20%. **h** Increased evidence for a Context representation ($P_{context}$) correlated with less EV/$EV_{back}$ competition (i.e., weaker effect of $P_{EV_{back}}^{2D}$ on $P_{EV}$ when $P_{context}$ was stronger, LR-test with interaction term: $p = 0.022$). Lines reflect model predictions, error bands represent 89% CI and vertical lines show group means of the top/bottom 20% of data (averaged first within participant, for individual lines, see Fig. S10). NB that $P_{context}$ was split into three levels for visualization; in our model it was continuous. **i** Comparing models of $P_{EV}$ (nested within $EV_{back}$ levels) revealed that adding either $P_{EV_{back}}^{2D}$ or $P_{context}$ improved model fit (**g** and **h**, $p = 0.063$ and $p = 0.022$), as well as their interaction $P_{context} \times P_{EV_{back}}^{2D}$ (LR-test with interaction compared to only $P_{context}$: $p = 0.022$, and only $P_{EV_{back}}^{2D}$: $p = 0.029$, $n = 35$). Note that $P_{EV_{back}}$ (**a**–**f**) indicates the Value classifier' class probabilities of the $EV_{back}$ class, whereas $P_{EV_{back}}^{2D}$ (**g**–**i**) indicates the $EV_{back}$ classifier's $EV_{back}$ class probabilities (the former was trained on 1D, the latter on 2D trials). Stars in (**c**), (**e**), (**f**), (**i**) represent threshold of $p < 0.05$. Source data are provided as a Source Data file.

Congruency of a given 2D trial. This analysis revealed that $EV_{back}$ had a negative effect on $P_{EV}$ (LR-test compared to null model $\chi^2_{(1)} = 5.96$, $p = 0.015$, Fig. 5b), meaning that larger irrelevant expected value led to weaker representation of the relevant one (measured by lower probability of the objective EV, $P_{EV}$). Importantly, this effect cannot be attributed to attentional effects caused by perceptual input, since replacing $EV_{back}$ with a regressor indicating the presence of its

corresponding perceptual feature in the training class, as highest or lowest value, did not provide a better model fit (AICs: −1229.2, −1223.3, respectively, see Fig. S8 for details). Adding the minimum value of the irrelevant context of the trial also did not improve the fit, indicating that it is specifically the highest of the two irrelevant features driving this effect (LR-test with added term: $\chi^2_{(1)} = 0.63$, $p = 0.43$). We found no evidence for a $EV_{back} \times P_{context}$ interaction (LR-test with added term:

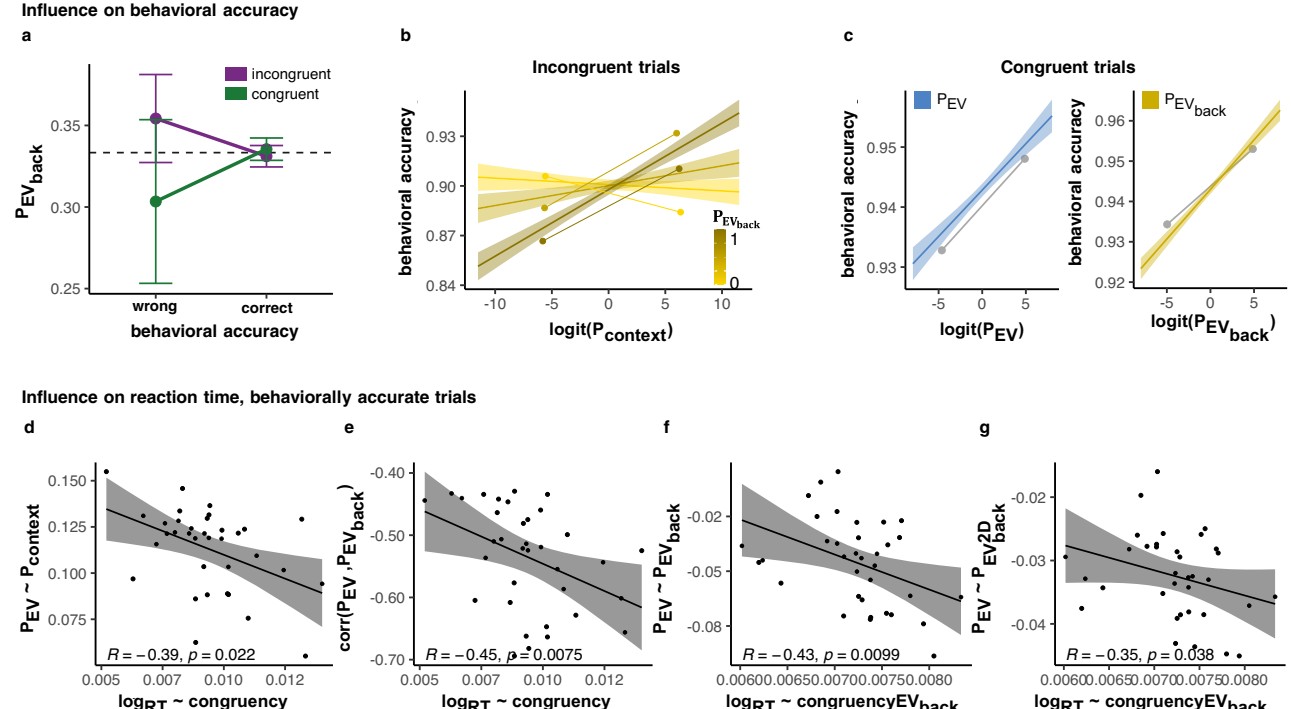

**Fig. 6 | vmPFC representations of context and value jointly guide behavior.**
**a**–**c** include all trials whereas **d**–**g** show only behaviorally accurate trials. **a** The Value classifier's probability of $EV_{back}$ ($P_{EV_{back}}$, $y$-axis) was increased when participants chose the option based on $EV_{back}$, corresponding to a wrong choice in incongruent trials (purple) and correct choice in congruent trials (green), LR-test vs. null model: $p = 0.034$, $n = 35$. Error bars represent corrected within subject SEMs[102,103].
**b** Decrease in behavioral accuracy ($y$-axis) in incongruent trials was marginally associated with lower context decodability (Context classifier, $x$-axis, $p = 0.051$). This effect was modulated by $EV_{back}$ representation, i.e., stronger in trials with higher $P_{EV_{back}}$ in vmPFC (shades of gold, $p = 0.012$, discretisation only for visualization). $p$ values represent LR-test with added terms and error bands represent the 89% CI. **c** Value classifier decodability of EV (blue, left) and $EV_{back}$ (gold, right) were both positively related to behavioral accuracy in congruent trials ($ps$: 0.058 and 0.009, respectively, $y$ axis). Lines are fitted slopes. Gray dots are group means of top and bottom 20% of data (within participant, for individual lines, see Fig. S11). $p$

values represent LR-test with added terms and error bands represent the 89% confidence interval. **d** Participants with weaker associations between Context and EV representations ($y$-axis, Fig. 5f), had a stronger Congruency RT effect ($x$-axis, larger values indicate stronger RT difference between incongruent and congruent trials, i.e., distance between purple and green lines in Fig. 2b). **e** More negative correlations between EV and $EV_{back}$ representations ($y$-axis, Fig. 4b) were associated with stronger Congruency RT effects ($x$-axis, see **d**). **f** Participants with a stronger (negative) link between $P_{EV}$ and $EV_{back}$ ($y$-axis, see Fig. 5e) also had a stronger $EV_{back}$ modulation on the Congruency RT effect ($x$-axis, see distance between purple and green lines in Fig. 2d). **g** Participants with a more negative link between $P_{EV}$ and $P_{EV_{back}}^{2D}$ ($y$-axis, more negative indicate stronger decrease, see Fig. 5g), had a stronger modulation of $EV_{back}$ on Congruency RT effect ($x$-axis, see **f**). **d**–**g** present Pearson correlations, $p$ values represent Spearman's $\rho$ statistic to estimate a rank-based measure of association[104,105] and error bands represent 95% confidence interval. Source data are provided as a Source Data file.

$\chi^2_{(1)} = 0.012$, $p = 0.91$). Our RSA analysis also provided further support for this effect, where we found that $EV_{back}$ also had a negative effect on the EV similarity, i.e., higher dissimilarity for higher $EV_{back}$ (Type II Wald $\chi^2$ test: $\chi^2_{(1)} = 36.6$, $p < 0.001$, see Fig. S6). Similarly, high $EV_{back}$ also disrupted the similarities between of the probabilities of the value classifier (LR-test: $\chi^2_{(1)} = 6.16$, $p = 0.013$, see Fig. S7). A number of control analyses also indicated the validity of finding: interestingly, and unlike in the behavioral models, we found that neither Congruency nor its interaction with EV or $EV_{back}$ influenced $P_{EV}$ ($\chi^2_{(1)} = 0.035$, $p = 0.852$, $\chi^2_{(1)} = 0.48$, $p = 0.787$, $\chi^2_{(1)} = 0.99$, $p = 0.317$, respectively, Fig. 5c), and a match of value expectations of both contexts (i.e., EV = $EV_{back}$) led no change of $P_{EV}$ ($\chi^2_{(1)} = 0.45$, $p = 0.502$, see "Methods"). We also found no effect of time since switch on the decodability of EV (Type II Wald $\chi^2$ test: $\chi^2_{(1)} = 0.85$, $p = 0.36$, Fig S9, but see discussion on limitations). Alternative models of $P_{EV}$, e.g., including within-option or between-context value differences, or alternatives for $EV_{back}$ (Fig. S9).

The decrease in value decodability due to high irrelevant value expectations could reflect a general disturbance of the value retrieval process caused by the distraction of competing values. Alternatively, the encoding of $EV_{back}$ could directly compete with the representation of EV—reflecting that the relevant and irrelevant value expectations might be represented using similar neural codes (note that the classifier was trained in the absence of task-irrelevant values, i.e., the

objective EV of 1D trials). In order to test this idea, we looked at the Value classifier probabilities in trials where EV ≠ $EV_{back}$. This allowed us to interpret the class probabilities of our Value classifier as either signifying EV ($P_{EV}$), $EV_{back}$ ($P_{EV_{back}}$) or a value that was expected in neither case ($P_{other}$, Fig. 5d). We then examined the correlation between each pair of classes. To prevent any disadvantage of the "other" class, we included only trials in which the "other" value's associated feature appeared on the screen (relevant or irrelevant). Note that the three class probabilities for each trial sum up to 1 and hence are strongly biased to correlate negatively. Yet, $P_{EV}$ and $P_{EV_{back}}$ had a significantly more negative correlation than $P_{EV}$ and $P_{other}$ ($\rho = -0.56$, $\sigma = 0.22$, $\rho = -0.40$, $\sigma = 0.25$ respectively, paired $t$-test: $t_{(34)} = -2.77$, $p = 0.017$, Fig. 5e). This shows that when the probability assigned to the EV decreased, it was accompanied by a stronger increase in the probability assigned to $EV_{back}$, akin to a competition between both types of expectations. Formally, we show that adding $P_{EV_{back}}$ to the model predicting $P_{EV}$ results in a smaller AIC than when adding $P_{other}$ (−574 vs. −473, respectively, Fig. 5f), likelihood-ratio-test for a model with $P_{EV_{back}}$: $\chi^2_{(1)} = 144.34$, $p < 0.001$, and with $P_{other}$: $\chi^2_{(1)} = 43.83$, $p < 0.001$).

The previous analysis only informs us about the overall correlation of probabilities across the entire experiment. To investigate the trial-wise dynamics of the neural representation within vmPFC, we trained an additional classifier to detect the $EV_{back}$ on behaviorally

accurate 2D trials. Although this classifier suffers from some caveats (see "Methods", Fig. S6a–c and below for details), we reasoned that trialwise probability fluctuations are unbiased, and proceeded to ask if the probability the $EV_{back}$ classifier assigned to the correct class ($P_{EV_{back}}^{2D}$) might relate to encoding of the relevant value as indicated by the Value classifier (i.e., $P_{EV}$). Importantly, both classifiers were trained on independent data ($EV_{back}$ classifier on 2D, and Value classifier on 1D trials), but in both cases on behaviorally accurate trials, i.e., trials where participants choose according to EV, as indicated by the relevant context. This model showed that an increase in neural representation of $EV_{back}$, when measured independently ($P_{EV_{back}}^{2D}$), reduced EV decodability on a trial-wise basis (lowered AIC score from −1223.6 to −1225.0, but note that in the LR-test $\chi_{(1)}^2 = 3.45$, $p = 0.063$, Fig. 5d). Most remarkably, the effect of Context, $P_{context}$, interacted with the effect of $P_{EV_{back}}^{2D}$, such that when the context signal was stronger, the negative effect of irrelevant value signals on relevant value signals was weaker (i.e., $P_{context}$ affected the association between $P_{EV_{back}}^{2D}$ and $P_{EV}$, LR-test: $\chi_{(1)}^2 = 5.22$, $p = 0.022$, Fig. 5e). In other words, the stronger the relationship between Context and EV representations, the less vmPFCs irrelevant value signal competed with its value representations, akin to a shielding effect. The same analysis also confirmed our previous finding that the strength of context encoding affected value encoding (effect of $P_{context}$, LR-test: $\chi_{(1)}^2 = 9.99$, $p = 0.002$). Note that the above analysis was complicated by the frequency differences between different $EV_{back}$ classes, which we controlled by running the model of $P_{EV}$ with random effects nested within levels of $EV_{back}$ for each subject, i.e., any effect found is not influenced by the (biased) mean difference between the probabilities assigned to each of those levels (intuitively, this is similar to running each correlation separately within each level of $EV_{back}$). Full models including effect sizes and confidence intervals can be found in Tables S3 and S4.

In summary, we showed the neural representation of EV was reduced in trials with higher expected value of the irrelevant context, and weakened EV representations were accompanied by an increase in neural representations of such irrelevant value expectation, in the same vmPFC region. The effect occurred irrespective of action-conflict between the relevant and irrelevant values (unlike participants' behavior). Most strikingly, the negative influence of $EV_{back}$ representation on EV decodability was mediated by a neural context signal, i.e., when the link between Context and EV increased, the effect of $EV_{back}$ representations diminished. As will be discussed later in detail, we consider this to be evidence for parallel processing of two task aspects in this region, EV and $EV_{back}$.

**Neural representation of EV, $EV_{back}$ and Context guide choice behavior.** Finally, we investigated how vmPFC's representations of EV, $EV_{back}$ and context influence participants' behavior. We first investigated this influence on choice accuracy. Note that the two contexts only indicate different choices in incongruent trials, where a wrong choice could be a result of a strong influence of the irrelevant context. Motivated by our behavioral analyses that indicated an influence of the irrelevant context on accuracy, we asked whether $P_{EV_{back}}$ was different on behaviorally wrong or incongruent trials. We found an interaction of accuracy × Congruency (Type II Wald $\chi^2$ test: $\chi_{(1)}^2 = 4.51$, $p = 0.034$, Fig. 6a) that indicated increases in $P_{EV_{back}}$ in accurate congruent trials and decreases in wrong incongruent trials. Hence, on trials in which participants erroneously chose the option with higher-valued irrelevant features, $P_{EV_{back}}$ was increased. Focusing only on behaviorally accurate trials, we found no effect of EV or Congruency on $P_{EV_{back}}$ (Type II Wald $\chi^2$ tests: $\chi_{(1)}^2 = 0.07$, $p = 0.794$, $\chi_{(1)}^2 = 0.00$, $p = 0.987$, respectively). This effect is preserved when modeling only wrong trials (Type II Wald $\chi^2$ test of Congruency: $\chi_{(1)}^2 = 4.36$, $p = 0.037$).

Motivated by the different predictions for congruent and incongruent trials, we next turned to model these trial-types separately. When focusing on incongruent trials we found that a weaker

representation of the relevant context was marginally associated with an increased error rate (negative effect of $P_{context}$) on accuracy, indicating an increased representation of the wrong context, LR-test: $\chi_{(1)}^2 = 3.66$, $p = 0.055$, Fig. 6b). Moreover, we found that the joint increases of the wrong context and its associated irrelevant expected value representation ($EV_{back}$) strengthened this effect, i.e., adding a $P_{context} × P_{EV_{back}}$ term to the model of error rates improved model fit (LR-test: $\chi_{(1)}^2 = 6.33$, $p = 0.012$, Fig. 6b; NB that we found no main effects of EV or $EV_{back}$ LR-tests: $\chi_{(1)}^2 = 0.28$, $p = 0.599$, $\chi_{(1)}^2 = 0.0$, $p = 0.957$, respectively). We next turned to congruent trials, where a wrong choice should not be associated with activation of the wrong context since both contexts indicate the same choice. Indeed, there was no influence of $P_{context}$ on accuracy in Congruent trials (LR-test: $\chi_{(1)}^2 = 0.0$, $p = 0.922$). However, strong representation of either relevant or irrelevant EV should lead to a correct choice. Indeed, we found that both an increase in $P_{EV_{back}}$ and (marginally) in $P_{EV}$ had a positive relation to behavioral accuracy ($\chi_{(1)}^2 = 3.5$, $p = 0.061$, $\chi_{(1)}^2 = 6.48$, $p = 0.011$, respectively, Fig. 6c).

Finally, we investigated reaction times of behaviorally accurate trials. In line with the results presented above, we found that participants who had a weaker influence of Context activity on their EV representation, also had a stronger RT Congruency effect ($r = -0.39$, $p = 0.022$, Fig. 6d). Next, we hypothesized that increased conflict between EV and $EV_{back}$ representations of should influence RT. Indeed, all neural signatures of $EV/EV_{back}$ conflict correlated with the Congruency-related RT effect: the more negative a participant's correlation between $P_{EV}$ and $P_{EV_{back}}$ was, the stronger her RT Congruency effect ($r = -0.45$, $p = 0.008$, Fig. 6e); a more negative association between $EV_{back}$ and $P_{EV}$ was linked to a stronger $EV_{back}$ modulation of the RT Congruency effect ($r = 0.43$, $p = 0.01$, Fig. 6f); finally, the same was true when considering the strength of the effect of the neural representation of $EV_{back}$ ($P_{EV_{back}}^{2D}$) on the neural EV signal in relation to the above behavioral marker ($r = 35$, $p = 0.004$, Fig. 6g). In other words, the negative influence of irrelevant EV and its neural representation on relevant EV signal, related to the interactive effect of $EV_{back} ×$ Congruency on RTs (i.e., slower RT for incongruent and faster for congruent trials).

In sum, choice accuracy was negatively related to the representation of irrelevant contexts and its associated value only in incongruent trials (i.e., when it mattered), while in congruent trials neural representations of EV and $EV_{back}$ contributed to accuracy. RT analyses showed that markers of (1) weaker representational link between context and EV and (2) stronger conflict between $EV_{back}$ and EV were both associated with a stronger influence of the counterfactual choice on their RT. Brought together these findings show that the representations of EV, $EV_{back}$ and Context in vmPFC do not only interact with each other, but guide choice behavior as reflected in accuracy as well as RT in behaviorally accurate trials.

**No univariate evidence for effects of irrelevant values on expected value signals in vmPFC.** The above analyses indicated that multiple value expectations are represented in parallel within vmPFC. Lastly, we asked whether whole-brain univariate analyses could also uncover evidence for processing of multiple value representations. Detailed description of the univariate analysis can be found in Fig. S12. Unlike the multivariate analysis, this revealed no positive modulation of Congruency, $EV_{back}$ or their interaction was observed in any frontal region. A negative effect of was found $EV_{back}$ in the Superior Temporal Gyrus, $p < 0.001$, Fig. S12c). We also found no region for the univariate effect of Congruency × $EV_{2D}$ interaction (even at $p < 0.005$). However, we found a negative univariate effect of Congruency × $EV_{back}$ in the primary motor cortex at a liberal threshold, which indicated that the difference between Incongruent and Congruent trials increased with higher $EV_{back}$, akin to a response conflict ($p < 0.005$, Fig. S12d). These findings contrast with the idea that competing values would have been

integrated into a single EV representation in vmPFC, because this account would have predicted a higher signal for Congruent compared to incongruent trials.

## Discussion

We investigated how contextually-irrelevant value expectations influence behavior and neural activation patterns in vmPFC. Participants reacted slower when the irrelevant context favored a different choice and faster when it favored the same. This Congruency effect increased with increasing reward associated with the hypothetical choice in the irrelevant context ($EV_{back}$). fMRI analyses of vmPFC voxels sensitive to the objective, i.e., relevant, expected value (EV) showed that (1) vmPFC contains a multifaceted representation of each trials expected value, irrelevant expected value and context; and that (2) higher irrelevant expected values, or a stronger neural representation of them, impaired the expected value signal, akin to a representational conflict between the two values. This conflict was moderated by the strength of the context signal, such that a stronger context signal was associated with a stronger expected value signal, and a diminished negative effect of the expected value of the irrelevant context. The different facets of vmPFC's representations were linked participants' behavior in a manner generally consistent with the idea that the representations of the alternative/irrelevant context and its associated value were present within vmPFC and guided behavior. The strength of these representations within vmPFC was related to slower and less accurate choices when the different contexts implied different actions, and faster and more accurate choices when they agreed on the action to be made.

One notable aspect of our experiment was that feature relevance was cued on each trial, and rewards were never influenced by irrelevant features. Nevertheless, participants' behavior was influenced by the expected outcome of the counterfactual choice. This supports the notion that cognitive control based arbitration between relevant and irrelevant features is incomplete[25,28,29]. Our neural analyses showed how internal value expectation(s) within vmPFC were shaped by such incomplete suppression: not the ignored context per se influenced vmPFC signals, but rather the computed expected value of the counterfactual choice that would have been made in that context. This was evidenced by the fact that the expected value of the background captured fluctuations in value representations. A control analysis showed that this cannot be explained by the presence of its corresponding perceptual-feature on the screen. Hence, our results cannot be explained by value-independent attention capture caused by the "distracting" irrelevant context (Fig. S8), and go beyond previous research on cognitive control, such as the Stroop Task[23].

We also asked whether relevant and irrelevant expected values integrate into a single EV, but found neither univariate nor multivariate evidence for this possibility. Specifically, we found no univariate $EV_{back}$ or congruency effects, and no increase in EV decodability when EV equalled $EV_{back}$. This suggests some differences in the underlying representations of relevant and irrelevant expected values. At the same time, our analysis showed that the value classifier was sensitive to the expected value of the irrelevant context in 2D trials, even though it was trained on 1D trials during which irrelevant values were not present. This suggests that within vmPFC "conventional" expected values and counterfactual values are encoded using partially, but not completely, similar patterns. Moreover, our results suggest that the EV of each context were activated simultaneously and competed with each other, a competition governed by the context signal. While neural evidence for EV competition did link behavioral evidence of choice conflict, we found no influence of action-congruency on vmPFC signal itself. This suggests that the conflicts between incongruent motor commands might be resolved elsewhere. Univariate analyses revealed that primary motor cortex was sensitive to Congruency, and hence might be the site of conflict resolution, in line with studies that suggest

distracting information can be found in task execution cortex in humans and monkeys[28,29]. The idea that the conflict between multiple values encoded in vmPFC is resolved in motor cortex and is also in line with our interpretation that vmPFC does not integrate both tasks into a single EV representation that drives choice.

Participants repeatedly had to switch between contexts in our task, a process that is well known to engage cognitive control mechanisms[22–25,32]. We evaluated to what extent this task switching affected our results and found that behavioral effects hold when excluding the first 2 trials after a context switch, and that the distance from the last switch did not interact with the influence of the irrelevant values (Fig. S2). Likewise, we found no influence of task switching on multivariate EV effects in vmPFC. Note, however, that due to our design we could not create balanced training sets (with respect to number of trials since context switch) which would be required for a more thorough investigation of the effect of trials since switch on value signals. We therefore conclude that while context switching is part of the investigated phenomenon, its presence alone cannot explain our findings.

Another important implication of our study concerns the nature of neural representations in vmPFC/mOFC, and in particular the relationship between state[35,37,38,49] and value[2–7] codes in this area. In order to compare both aspects, we used a categorical classifier for value as well as states, rather than examining continuous value representations. Nevertheless, we believe that the value similarity analysis (both in the RSA, Fig. 3d, e and classifier probabilities, Fig. S7) additionally shows evidence for such continuous value representations. We specifically chose to focus on the vmPFC region that is commonly investigated in value-based decision research. We therefore defined our ROI in a univariate manner as commonly done in the literature (e.g. refs. 4,5) and studied the multivariate state and value signal within this ROI (e.g. refs. 35,37). We found that in addition to (expected) value information, vmPFC/mOFC also represented the context or task-state, which identified relevant information and thereby disambiguated the partially observable sensory state (e.g. refs. 35,37,49). Note that in our case the task context was agnostic to value (which was balanced across contexts) and specific features, but rather consisted of a superset of the more specific motion direction/color features. Any area sensitive to these more specific states would therefore also show decoding of context as defined here. Another methodological aspect was that we decoded based on timeshifted TR images, rather than deconvolved activity patterns[50] as is common practice in fMRI decoding papers[18,51–53]. Decoding level and approach may have implications for the representations that can be uncovered in future research. Overall, our findings are in line with work that has found that EV could be one additional aspect of OFC activity[44], which is multiplexed with other task-related information. Crucially, the idea that state representations integrate different kinds of task-relevant information[40,54] could explain why this region was found to be crucial for integrating valued features when all features of an object are relevant for choice[19,40], although some work suggests that it might also reflect integration of features not carrying any value[41].

To conclude, the main contribution of our study is that we elucidated the relation between task-context and value representations within vmPFC. By introducing multiple possible values of the same option in different contexts, we were able to reveal a complex representation of task structure in vmPFC, with both task-contexts and their associated expected values activated in parallel. The decodability of both contexts and EVs independently from vmPFC, and their relation to choice behavior, hints at integrated computation of these in this region. We believe that this bridges between findings of EV representation in this region to the functional role of this region as representing task-states, whereby relevant and counterfactual values can be considered as part of a more encompassing state representation.

## Methods

The study complies with all relevant ethical regulations and was approved by the ethics board of the Free University Berlin (Ref. Number: 218/2018).

### Participants

Forty right-handed young adults took part in the experiment (18 women, $\mu_{age} = 27.6$, $\sigma_{age} = 3.35$) in exchange for monetary reimbursement. Participants were recruited using the participant database of Max-Planck-Institute for Human Development. Beyond common MRI-safety related exclusion criteria (e.g., piercings, pregnancy, large or circular tattoos etc.), we also did not admit participants to the study if they reported any history of neurological disorders, tendency for back pain, color perception deficiencies or if they had a head circumference larger than 58 cm (due to the limited size of the 32-channel head-coil). Gender of participants was self-reported (note that the study was conducted in the German language where there is no clear distinction between sex and gender). We had no reason to suspect any gender differences in the task and therefore did not include this information in the analyses. After data acquisition, we excluded five participants from the analysis; one for severe signal drop in the OFC, i.e., more than 15% less voxels in functional data compared to the OFC mask extracted from freesurfer parcellation of the T1 image[55,56]. One participant was excluded due to excessive motion during fMRI scanning (more than 2 mm in any axial direction) and three participants for low performance (<75% accuracy in one context in the main task). In the behavioral-replication, 23 young adults took part (15 women, $\mu_{age} = 27.1$, $\sigma_{age} = 4.91$) and two were excluded for the same accuracy threshold. Due to technical reasons, 3 trials (4 in the replication sample) were excluded since answers were recorded before stimulus was presented and 2 trials (non in the replication) in which RT was faster than 3 SD from the mean (likely premature response). The monetary reimbursement consisted of a base payment of 10 Euro per hour (8.5 for replication sample) plus a performance dependent bonus of 5 Euro on average.

### Experimental procedures

**Design.** Participants performed a random dot-motion paradigm in two phases, separated by a short break (minimum 15 min). In the first phase, psychophysical properties of four colors and four motion directions were first titrated using a staircasing task. Then, participants learned the rewards associated with each of these eight features during a outcome learning task. The second phase took place in the MRI scanner and consisted mainly of the main task, in which participants were asked to make decisions between two random dot kinematograms, each of which had one color and/or one direction from the same set. Note there were two additional mini-blocks of 1D trials only, at the end of first- and at the start of the second phase (during anatomical scan, see below). The replication sample completed the same procedure with the same break length, but without MRI scanning. That is, both phases were completed in a behavioral testing room. Details of each task and the stimuli are described below. Behavioral data were recorded during all experiment phases. MRI data were recorded during phase 2. We additionally collected eye-tracking data (EyeLink 1000; SR Research Ltd.; Ottawa, Canada) both during the staircasing and the main decision making task to ensure continued fixation (data not presented). The overall experiment lasted between 3.5 and 4 h (including the break between the phases). Additional information about the pre-scanning phase can be found in Fig. S1.

**Room, luminance and apparatus.** Behavioral sessions were conducted in a dimly lit room without natural light sources, such that light fluctuations could not influence the perception of the features. A small lamp was stationed in the corner of the room, positioned so it would not cast shadows on the screen. The lamp had a light bulb with 100% color rendering index, i.e., avoiding any influence on color perception. Participants sat on a height adjustable chair at a distance of 60 cm from a 52 cm horizontally wide, Dell monitor (resolution: 1920 × 1200, refresh rate 1/60 frames per second). Distance from the monitor was fixed using a chin-rest with a head-bar. Stimuli were presented using psychtoolbox version 3.0.11[57–59] in MATLAB R2017b[60]. In the MRI-scanner room lights were switched off and light sources in the operating room were covered in order to prevent interference with color perception or shadows cast on the screen. Participants lay inside the scanner at distance of 91 cm from a 27 cm horizontally wide screen on which the task was presented a D-ILA JVC projector (D-ILa Projektor SXGA, resolution: 1024 × 768, refresh rate: 1/60 frames per second). Stimuli were presented using psychtoolbox version 3.0.11[57–59] in MATLAB R2017b[60] on a Dell precision T3500 computer running windows XP version 2002.

**Stimuli.** Each cloud of dots was presented on the screen in a circular array with 7° visual angle in diameter. In all trials involving two clouds, the clouds appeared with 4° visual angle distance between them, including a fixation circle (2° diameter) in the middle, resulting in a total of 18° field of view (following total apparatus size from ref. 46). Each cloud consisted of 48 square dots of 3 × 3 pixels. We used four specific motion and four specific color features.

To prevent any bias resulting from the correspondence between response side and dot motion, each of the four motion features was constructed of two angular directions rotated by 180°, such that motion features reflected an axis of motion, rather than a direction. Specifically, we used the four combinations: 0°–180° (left–right), 45°–225° (bottom right to upper left), 90°–270° (up-down) and 135°–315° (bottom left–upper right). We used a Brownian motion algorithm (e.g. ref. 46), meaning in each frame a different set of given amount of coherent dots was chosen to move coherently in the designated directions in a fixed speed, while the remaining dots moved in a random direction (Fig. S1). Dots speed was set to 5° per second (i.e., 2/3 of the aperture diameter per second, following[46]). Dots lifetime was not limited. When a dot reached the end of the aperture space, it was sent "back to start", i.e., back to the other end of the aperture. Crucially, the number of coherent dots (henceforth: motion-coherence) was adjusted for each participant throughout the staircasing procedure, starting at 0.7 to ensure high accuracy (see ref. 46). An additional type of motion-direction was "random-motion" and was used in 1D color clouds. In these clouds, dots were split to four groups of 12, each assigned with one of the four motion features and their adjusted-coherence level, resulting in a balanced subject-specific representation of random motion.

In order to keep the luminance fixed, all colors presented in the experiment were taken from the YCbCr color space with a fixed luminance of $Y = 0.5$. YCbCr is believed to represent human perception in a relatively accurate manner (cf. [61]). In order to generate an adjustable parameter for the purpose of staircasing, we simulated a squared slice of the space for $Y = 0.5$ (Fig. S1) in which the representation of the dots color moved using a Brownian motion algorithm as well. Specifically, all dots started close to the (gray) middle of the color space, in each frame a different set of 30% of dots was chosen to move coherently toward the target color in a certain speed whereas all the rest were assigned with a random direction. Perceptually, this resulted in all the dots being gray at the start of the trial and slowly taking on the designated color. Starting point for each color was chosen based on pilot studies and was set to a distance of 0.03–0.05 units in color space from the middle. Initial speed in color space (henceforth: color-speed) was set so the dots arrive to their target (23.75% the distance to the corner from the center) by the end of the stimulus presentation (1.6 s). i.e., distance to target divided by the number of frames per trial duration. Color-speed was adjusted throughout the staircasing

procedure. An additional type of color was "no color" for motion 1D trials for which we used the gray middle of the color space.

**Staircasing task.** In order to ensure RTs mainly depended on associated values and not on other stimulus properties (e.g., salience), we created a staircasing procedure that was conducted prior to value learning. In this procedure, motion-coherence and color-speed were adjusted for each participant in order to minimize between-feature detection time differences. As can be seen in Fig. S1, in this perceptual detection task participants were cued (0.5 s) with either a small arrow (length 2°) or a small colored circle (0.5° diameter) to indicate which motion-direction or color they should choose in the upcoming decision. After a short gray (middle of YCbCr) fixation circle (1.5 s, diameter 0.5°), participants made a decision between the two clouds (1.6 s). Clouds in this part could be either both single-feature or both dual-features. In dual feature trials, each stimulus had one color and one motion feature, but the cue indicated either a specific motion or a specific color. After a choice, participants received feedback (0.4 s) whether they were (1) correct and faster than 1 s, (b) correct and slower or (c) wrong. After a short fixation (0.4 s), another trial started. All timings were fixed in this part. Participants were instructed to always look at the fixation circle in the middle of the screen throughout this and all subsequent tasks. To motivate participants and continued perceptual improvements during the later (reward related) task-stages, participants were told that if they were correct and faster than 1 s in at least 80% of the trials, they will receive an additional monetary bonus of 2 Euros.

The staircasing started after a short training (choosing correct in 8 out of 12 consecutive trials mixed of both contexts) and consisted of two parts: two adjustment blocks an two measurement blocks. All adjustments of color-speed and motion-coherence followed this formula:

$$\theta_i^{t+1} = \theta_i^t + \alpha\theta_i^t \frac{\overline{RT_i^t} - RT^0}{RT^0} \tag{1}$$

where $\theta_i^{t+1}$ represents the new coherence/speed for motion or color feature $i$ during the upcoming time interval/block $t+1$, $\theta_i^t$ is the level at the time of adjustment, $\overline{RT_i^t}$ is the mean RT for the specific feature $i$ during time interval $t$, $RT_0$ is the "anchor" RT toward which the adjustment is made and $\alpha$ represents a step size of the adjustment, which changed over time as described below.

The basic building block of adjustment blocks consisted of 24 cued-feature choices for each context ($4 \times 3 \times 2 = 24$, i.e., 4 colors, each discriminated against 3 other colors, on 2 sides of screen). The same feature was not cued more than twice in a row. Due to time constrains, we could not include all possible feature-pairing combinations between the cued and uncued features. We therefore pseudo-randomly choose from all possible background combinations for each feature choice (unlike later stages, this procedure was validated on and therefore included also trials with identical background features). In the first adjustment block, participants completed 72 trials, i.e., 36 color-cued and 36 motion-cued, interleaved in chunks of 4–6 trials in a non-predictive manner. This included, for each context, a mixture of one building block of 2D trials and half a block of 1D trials, balanced to include 3 trials for each cued-feature. 1D or 2D trials did not repeat more than three times in a row. At the end of the first adjustment block, the mean RT of the last 48 (accurate) trials was taken as the anchor ($RT^0$) and each individual feature was adjusted using the above formula with $\alpha = 1$. The second adjustment block started with 24 motion-cued only trials which were used to compute a new anchor. Then, throughout a series of 144 trials (72 motion-cued followed by 72 color-cued trials, all 2D), every three correct answers for the same feature resulted in an adjustment step for that specific feature (Eq. (1)) using the average RT of these trials ($\overline{RT_i^t}$) and the motion anchor $RT^0$

for both contexts. This resulted in a maximum of six adjustment steps per feature, where alpha decreased from 0.6 to 0.1 in steps of 0.1 to prevent over-adjustment.

Next, participants completed two measurement blocks identical in structure to the main task (see below) with two exceptions: first, although this was prior to learning the values, they were perceptually cued to chose the feature that later would be assigned with the highest value. Second, to keep the relevance of the feature that later would take the lowest value (i.e., would rarely be chosen), we added 36 additional trials cued to choose that feature (18 motion and 18 color trials per block).

**Outcome learning task.** After the staircasing and prior to the main task, participants learned to associate each feature with a deterministic outcome. Outcomes associated with the four features on each contexts were 10, 30, 50 and 70 credit-points. The value mapping to perceptual features was assigned randomly between participants, such that all possible color- and all possible motion-combinations were used at least once ($4! = 24$ combinations per context). We excluded motion value-mapping that correspond to clockwise or counter-clockwise ordering. The outcome learning task consisted only of single-feature clouds, i.e., clouds without coherent motion or dots "without" color (gray). Therefore each cloud in this part only represented a single feature. To encourage mapping of the values for each context on similar scales, the two clouds could be either of the same context (e.g., color and color) or from different contexts (e.g., color and motion). Such context-mixed trials did not repeat in other parts of the experiment.

The first block of the outcome learning task had 80 forced choice trials (5 repetitions of 16 trials: 4 values × 2 Context × 2 sides of screen), in which only one cloud was presented, but participants still had to choose it to observe its associated reward. These were followed by mixed blocks of 72 trials which included 16 forced choice interleaved with 48 free choice trials between two 1D clouds (6 value-choices: 10 vs. 30/50/70, 30 vs. 50/70, 50 vs. 70 × 4 context combinations × 2 sides of screen for highest value). To balance the frequencies with which feature-outcome pairs would be chosen, we added eight forced choice trials in which choosing the lowest value was required. Trials were pseudo-randomized so no value would repeat more than three times on the same side and same side would not be chosen more the three consecutive times. Mixed blocks repeated until participants reached at least 85% accuracy of choosing the higher-valued cloud in a block, with a minimum of two and a maximum of four blocks. Since all clouds were 1D and choice could be between contexts, these trials started without a cue, directly with the presentation of two 1D clouds (1.6 s). Participants then made a choice, and after short fixation (0.2 s) were presented with the value of both chosen and unchosen clouds (0.4 s, with value of choice marked with a square around it, see Fig. S1). After another short fixation (0.4 s) the next trial started. Participants did not collect reward points in this stage, but were told that better learning of the associations will result in more points, and therefore more money later. Specifically, in the MRI experiment participants were instructed that credit points during the main task will be converted into a monetary bonus such that every 600 points they will receive 1 Euro at the end. The behavioral replication cohort received 1 Euro for every 850 points.

**Main task preparation.** In preparation of the main task, participants performed one block of 1D trials at the end of phase 1 and then at the start of the MRI session during the anatomical scan. These blocks were included to validate that changing presentation mediums between phases (computer screen vs. projector) did not introduce a perceptual bias to any features and as a final correction for post value-learning RT differences between contexts. Each block consisted of 30 color and 30 motion 1D trials interleaved in chunks of 4–7 trials in a non-predictive manner. The value difference between the clouds was fixed to 20

points (10 repetitions of 3 value comparisons × 2 contexts). Trials were pseudo-randomized so no target value was repeated more than once within context (i.e., not more than twice all in all) and was not presented on the same side of screen more than 3 consecutive trials within context and 4 in total. In each trial, they were first presented with a contextual cue (0.6 s) for the trial, followed by short fixation (0.5 s) and the presentation of two single-feature clouds of the cued context (1.6 s) and had to choose the highest valued cloud. After a short fixation (0.4 s), participants were presented with the chosen cloud's outcome (0.4 s). The timing of the trials was fixed and shorter than in the remaining main task because no functional MRI data was acquired during these blocks. Participants were instructed that from the first preparation block they started to collect the rewards. Data from these 1D block were used to inspect and adjust for potential differences between the MRI and the behavior setup. First, participants reacted generally slower in the scanner ($t(239) = -9.415$, $p < 0.001$, paired $t$-test per subject per feature). Importantly, however, we confirmed that this slowing was uniform across features, i.e., no evidence was found for a specific feature having more RT increase than the rest (ANOVA test on the difference between the phases, $F(7,232) = 1.007$, $p = 0.427$). Second, because pilot data indicated increased RT differences between contexts after the outcome learning task we took the mean RT difference between color and motion trials in the second mini-block in units of frames (RT difference divided by the refresh rate), and moved the starting point of each color relative to their target color, the number of frames × its speed. Crucially, the direction of the move (closer/further to target) was the same for all colors, thus ensuring not to induce within-context RT differences.

**Main task.** Finally, participants began with the main experiment inside the scanner. Participants were asked to choose the higher-valued of two simultaneously presented random dot kinematograms, based on the previously learned feature-outcome associations. As described in the main text, each trial started with a cue that indicated the current task context (color or motion). In addition, both clouds could either have two features (each a color and a motion, 2D trials) or one feature only from the cued context (e.g., colored, but randomly moving dots).

The main task consisted of four blocks in which 1D and 2D trial were intermixed. Each block contained 36 1D trials (3 EV × 2 Contexts × 6 repetitions) and 72 2D trials (3 EV × 2 Contexts × 12 feature-combinations, see Fig. 1c). Since this task took part in the MRI, the duration of the fixation circles were drawn from a truncated exponential distribution with a mean of $\mu = 0.6$ s (range 0.5–2.5 s) for the interval between cue and stimulus, a mean of $\mu = 3.4$ s (1.5–9 s) for the interval between stimulus and outcome and a mean of $\mu = 1.25$ s (0.7–6 s) for the interval between outcome and the cue of the next trial. The cue, stimulus and outcome were presented for 0.6, 1.6 and 0.8 s, respectively. Timing was optimized using VIF-calculations of trial-wise regression models (see "Classification procedure" section below).

The order of trials within blocks was controlled as follows: the cued context stayed the same for 4–7 trials (in a non-predictive manner), to prevent context confusion caused by frequent switching. No more than 3 repetitions of 1D or 2D trials within each context could occur, and no more than 5 repetition overall. The target did not appear on the same side of the screen on more than 4 consecutive trials. Congruent or incongruent trials did not repeat more than 3 times in a row. In order to avoid repetition suppression, i.e., a decrease in the fMRI signal due to a repetition of information (e.g. refs. 62,63), no target feature was repeated two trials in a row, meaning the EV could repeat maximum once (i.e., one color and one motion). As an additional control over repetition, we generated 1000 designs according the above-mentioned rules and choose the designs in which the target value was repeated in no more than 10% of trials across trial types, as well as when considering congruent, incongruent or 1D trials separately.

In all mixed effect models, When describing main effects of models, the $\chi^2$ represents Type II Wald $\chi^2$ tests, whereas when describing model comparison, the $\chi^2$ represents the log-likelihood ratio test. Model comparison throughout the paper was done using the "anova" function. The reason we used $\chi^2$ test is that classification probabilities as well as RSA dissimilarities are not normally distributed (these follow beta and gamma distributions respectively, note that the glmmTMB toolbox also uses $\chi^2$ as its default for these distributions). Regressors were scaled prior to fitting the models for all analyses.

Throughout the behavioral and fMRI analyses we report exact $p$ values unless they fall below 0.001, in which case we report $p < 0.001$.

**Behavioral analysis**

RT data was analyzed in R (R version 3.6.3[64], RStudio version 1.3.959[65]) using linear mixed effect models (lmer in lme4 1.1-21: ref. 66). The behavioral model that we found to fit the behavioral RT data best was:

$$\log \text{RT}_k^t = \beta_0 + \gamma_{0k} + \beta_1 \text{EV} + \beta_2 \text{Congruency}_t + \beta_3 \text{Congruency}_t \text{EV}_{\text{back}_t} \\ + \beta_4 \text{Congruency}_t \text{EV}_t + \nu_1 t + \nu_2 \text{side}_t + \nu_3 \text{switch}_t + \nu_4 \text{context}_t \tag{2}$$

where $\log \text{RT}_k^t$ is the log reaction time of subject $k$ in trial $t$, $\beta_0$ and $\gamma_{0k}$ represent global and subject-specific intercepts, $v$-coefficients reflect nuisance regressors (side of target object, trials since last context switch and the current context), $\beta_1$ to $\beta_4$ captured the fixed effect of EV, Congruency, Congruency × EV$_{\text{back}}$ and Congruency × EV, respectively. The additional models reported in the SI included intercept terms specific for each factor level, nested within subject (for EV, Block and Context, see Fig. S2). An exploratory analysis investigating all possible two-way interactions with all nuisance regressors can be found in Fig. S4.

Investigations of alternative parametrizations of the values can be found in Fig. S3.

Accuracy data were analyzed in R (R version 3.6.3[64], RStudio version 1.3.959[65]) using generalized linear mixed effect models (glmer in lme4 1.1-21: ref. 66) employing a binomial distribution family with a "logit" link function. Regressors were scaled prior to fitting the models for all analyses. No-answer trials of were excluded from this analysis. The model found to fit the behavioral accuracy data best was almost equivalent to the RT model, except for the fourth term involving Congruency × switch:

$$\text{ACC}_k^t = \beta_0 + \gamma_{0k} + \beta_1 \text{EV} + \beta_2 \text{Congruency}_t + \beta_3 \text{Congruency}_t \text{EV}_{\text{back}_t} \\ + \beta_4 \text{Congruency}_t \text{switch}_t + \nu_1 t + \nu_2 \text{side}_t + \nu_3 \text{switch}_t + \nu_4 \text{context}_t \tag{3}$$

where $\text{ACC}_k^t$ is the accuracy (1 for correct and 0 for incorrect) of subject $k$ in trial $t$ and all the rest of the regressors are equivalent to Eq. (2). An exploratory analysis investigating all possible two-way interactions with all nuisance regressors can be found in Fig. S5. We note that the interaction Congruency × switch indicates that participants were more accurate the further they were from a context switch point. Out of the nuisance variables, only "switch" influenced accuracy, Type II Wald $\chi^2$ test in baseline model: $\chi^2_{(1)} = 10.22$, $p = 0.001$.

**fMRI data**

**fMRI data acquisition.** MRI data was acquired using a 32-channel head coil on a research-dedicated 3-Tesla Siemens Magnetom TrioTim MRI scanner (Siemens, Erlangen, Germany) located at the Max Planck Institute for Human Development in Berlin, Germany. High-resolution T1-weighted (T1w) anatomical Magnetization Prepared Rapid Gradient Echo (MPRAGE) sequences were obtained from each participant to allow registration and brain surface reconstruction (sequence specification: 256 slices; TR = 1900 ms; TE = 2.52 ms; FA = 9 degrees; inversion time (TI) = 900 ms; matrix size = 192 × 256; FOV = 192 × 256 mm; voxel size = 1 × 1 × 1 mm). This was followed with two short acquisitions with six volumes each that were collected using the same sequence

parameters as for the functional scans but with varying phase encoding polarities, resulting in pairs of images with distortions going in opposite directions between the two acquisitions (also known as the blip-up/blip-down technique). From these pairs the displacements were estimated and used to correct for geometric distortions due to susceptibility-induced field inhomogeneities as implemented in the fMRIPrep preprocessing pipeline. In addition, a whole-brain spoiled gradient recalled (GR) field map with dual echo-time images (sequence specification: 36 slices; A-P phase encoding direction; TR = 400 ms; TE1 = 4.92 ms; TE2 = 7.38 ms; FA = 60 degrees; matrix size = 64 × 64; 619 FOV = 192 × 192 mm; voxel size = 3 × 3 × 3.75 mm) was obtained as a potential alternative to the method described above. However, this GR field map was not used in the preprocessing pipeline. Lastly, four functional runs using a multi-band sequence (sequence specification: 64 slices in interleaved ascending order; anterior-to-posterior (A-P) phase encoding direction; TR = 1250 ms; echo time (TE) = 26 ms; voxel size = 2 × 2 × 2 mm; matrix = 96 × 96; field of view (FOV) = 192 × 192 mm; flip angle (FA) = 71 degrees; distance factor = 0, MB acceleration factor = 4). A tilt angle of −30 degrees from AC-PC (tilted backwards, or: front side of FOV upwards) was used in order to maximize signal from the orbitofrontal cortex (OFC, see ref. 67). For each functional run, the task began after the acquisition of the first four volumes (i.e., after 5.00 s) to avoid partial saturation effects and allow for scanner equilibrium. Each run was about 15 min in length, including a 20 s break in the middle of the block (while the scanner is running) to allow participants a short break. We measured respiration and pulse during each scanning session using pulse oximetry and a pneumatic respiration belt part of the Siemens Physiological Measurement Unit. Full details of the sequences used, as provided by the MRI scanner, are shared in the same repository with the code (see "MRI_Sequences.pdf").

**BIDS conversion and defacing.** Data was arranged according to the brain imaging data structure (BIDS) specification[68] using the HeuDiConv tool (version 0.6.0.dev1; freely available from https://github.com/nipy/heudiconv). Dicoms were converted to the NIfTI-1 format using dcm2niix [version 1.0.20190410 GCC6.3.0; ref. 69]. In order to make identification of study participants highly unlikely, we eliminated facial features from all high-resolution structural images using pydeface (version 2.0; available from https://github.com/poldracklab/pydeface). The data quality of all functional and structural acquisitions were evaluated using the automated quality assessment tool MRIQC (for details, (see ref. 70), and the MRIQC documentation). The visual group-level reports confirmed that the overall MRI signal quality was consistent across participants and runs.

**fMRI preprocessing.** Data were preprocessed using fMRIPrep 1.2.6 (refs. 71,72; RRID:SCR_016216), which is based on Nipype 1.1.7 (refs. 73,74; RRID:SCR_002502). Many internal operations of fMRIPrep use Nilearn 0.5.0 (ref. 75, RRID:SCR_001362], mostly within the functional processing workflow.

Specifically, the T1-weighted (T1w) image was corrected for intensity non-uniformity (INU) using `N4BiasFieldCorrection` (ref. 76, ANTs 2.2.0), and used as a T1w-reference throughout the workflow. The anatomical image was skull-stripped using `antsBrainExtraction.sh` (ANTs 2.2.0), using OASIS as the target template. Brain surfaces were reconstructed using `recon-all` (FreeSurfer 6.0.1, RRID:SCR_001847, ref. 55), and the brain masks were estimated previously was refined with a custom variation of the method to reconcile ANTs-derived and FreeSurfer-derived segmentations of the cortical gray-matter of Mindboggle (RRID:SCR_002438, ref. 54). Spatial normalization to the ICBM 152 Nonlinear Asymmetrical template version 2009c (ref. 77, RRID:SCR_008796) was performed through nonlinear registration with `antsRegistration` (ANTs 2.2.0, RRID:SCR_004757, ref. 78), using brain-extracted versions of both T1w volume and

template. Brain tissue segmentation of cerebrospinal fluid (CSF), white-matter (WM) and gray-matter (GM) was performed on the brain-extracted T1w using `fast` (FSL 5.0.9, RRID:SCR_002823, ref. 79).

To preprocess the functional data, a reference volume for each run and its skull-stripped version were generated using a custom methodology of fMRIPrep. A deformation field to correct for susceptibility distortions was estimated based on two echo-planar imaging (EPI) references with opposing phase-encoding directions, using `3dQwarp`[80] (AFNI 20160207). Based on the estimated susceptibility distortion, an unwarped BOLD reference was calculated for a more accurate co-registration with the anatomical reference. The BOLD reference was then co-registered to the T1w reference using `bbregister` (FreeSurfer), which implements boundary-based registration[81]. Co-registration was configured with nine degrees of freedom to account for distortions remaining in the BOLD reference. Head-motion parameters with respect to the BOLD reference (transformation matrices, and six corresponding rotation and translation parameters) are estimated before any spatiotemporal filtering using `mcflirt` (FSL 5.0.9[82]). BOLD runs were slice-time corrected using `3dTshift` from AFNI 20160207 (ref. 80, RRID:SCR_005927) and aligned to the middle of each TR. The BOLD time-series (including slice-timing correction) were resampled onto their original, native space by applying a single, composite transform to correct for head-motion and susceptibility distortions. First, a reference volume and its skull-stripped version were generated using a custom methodology of fMRIPrep.

Several confound regressors were calculated during preprocessing: six head-motion estimates (see above), Framewise displacement, six anatomical component-based noise correction components (aCompCorr) and 18 physiological parameters (8 respiratory, 6 heart rate and 4 of their interaction). The head-motion estimates were calculated during motion correction (see above). Framewise displacement was calculated for each functional run, using the implementations in Nipype (following the definitions by ref. 83). A set of physiological regressors were extracted to allow for component-based noise correction (CompCor[84]). Principal components are estimated after high-pass filtering the BOLD time-series (using a discrete cosine filter with 128s cut-off) for the two CompCor variants: temporal (tCompCor, unused) and anatomical (aCompCor). For aCompCor, six components are calculated within the intersection of the aforementioned mask and the union of CSF and WM masks calculated in T1w space, after their projection to the native space of each functional run (using the inverse BOLD-to-T1w transformation). All resamplings can be performed with a single interpolation step by composing all the pertinent transformations (i.e., head-motion transform matrices, susceptibility distortion correction, and co-registrations to anatomical and template spaces). Gridded (volumetric) resamplings were performed using `antsApplyTransforms` (ANTs), configured with Lanczos interpolation to minimize the smoothing effects of other kernels[85]. Lastly, for the 18 physiological parameters, correction for physiological noise was performed via RETROICOR[86,87] using Fourier expansions of different order for the estimated phases of cardiac pulsation (3rd order), respiration (4th order) and cardio-respiratory interactions (1st order)[88]: the corresponding confound regressors were created using the Matlab PhysIO Toolbox (ref. 89, open source code available as part of the TAPAS software collection (Version 3.2.0): https://www.translationalneuromodeling.org/tapas. For more details of the pipeline, and details on other confounds generated but not used in our analyses, see https://fmriprep.readthedocs.io/en/latest/workflows.html the section corresponding to workflows in fMRIPrep's documentation.

For univariate analyses, BOLD time-series were re-sampled to MNI152NLin2009cAsym standard space in the fMRIPrep pipeline and then smoothed using SPM (ref. 90, SPM12 (7771)) with 8 mm FWHM, except for ROI generation, where a 4 mm FWHM kernel was used. Multivariate analyses were conducted in native space, and data was

smoothed with 4 mm FWHM using SPM (ref. 90, SPM12 (7771)). Classification analyses further involved three preprocessing steps of voxel time-series: First, extreme-values more than 8 standard deviations from a voxels mean were corrected by moving them by 50% their distance from the mean toward the mean (this was done to not bias the last $z$ scoring step). Second, the time-series of each voxel was detrended, a high-pass filter at 128 Hz was applied and confounds were regressed out in one action using Nilearn 0.6.2 (later changed to 0.7.0)[75]. Lastly, the time-series of each voxel for each block was $z$ scored.

## Univariate fMRI analysis

All GLMs were conducted using SPM12[90],SPM12 (7771) in MATLAB[60]. All GLMs consisted of two regressors of interest corresponding to the onsets of the two trial-types (1D/2D, except for one GLM where 2D onsets were split by Congruency) and included one parametric modulator of EV assigned to 1D onset and different combinations of parametric modulators of EV, Congruency, $EV_{back}$ and their interactions (see Fig. S13 for GLM visualization). All parametric modulators were demeaned before entering the GLM, but not orthogonalized. Regressors of no interest reflected cue onsets in Motion and Color trials, stimulus onsets in wrong and no-answer trials, outcome onsets and 31 nuisance regressors (e.g., motion and physiological parameters, see fMRI-preprocessing). The duration of stimulus regressors corresponded to the time the stimuli were on screen. The durations for the rest of the onset regressors were set to 0. Microtime resolution was set to 16 (64 slices/4 MB factor) and microtime onset was set to the 8 (since slice time correction aligned to middle slice, see fMRI-preprocessing). Data for all univariate analyses were masked with a whole brain mask computed as intercept of each functional run mask generated from fMRIprep[55,56]. MNI coordinates were translated to their corresponding brain regions using the automated anatomical parcellation toolbox (refs. 91–93, AAL3v1) for SPM. We verified the estimability of the design matrices by assessing the Variance Inflation Factor (VIF) for each onset regressor in the HRF-convolved design matrix. Specifically, for each subject, we computed the VIF (assisted by scripts from https://github.com/sjgershm/ccnl-fmri) for each regressor in the HRF-convolved design matrix and averaged the VIFs of corresponding onsets across the blocks. None of the VIFs surpassed a value of 3.5 (a value of 5 is considered a conservative indicator for overly colinear regressors, e.g. ref. 94, see Fig. S13 for details). Detailed descriptions of all GLMs are reported in the main text. Additional GLMs verifying the lack of Congruency in any frontal region can be found in Fig. S13.

## Functionally defined vmPFC ROI.

Our fMRI analyses focused on understanding the representations of expected values in vmPFC. We therefore first sought to identify a value-sensitive region of interest (ROI) that reflected expected values in 1D and 2D trials, following common procedures in the literature (e.g. ref. 4). We analyzed the fMRI data using general linear models (GLMs) with separate onsets and EV parametric modulators for 1D and 2D trials (at stimulus presentation with 0 s duration) and defined a functional ROI for value representations centered on vmPFC using the union of the EV modulators for 1D and 2D trials ($EV_{1D} + EV_{2D} > 0$), Fig. 3a, $p < 0.0005$ FDR corrected). Note that this GLM had no information regarding the contextually irrelevant context. The group ROI was generated in MNI space and included 998 voxels. Multivariate analyses were conducted in native space and the ROI was transformed to native space using ANTs and nearest neighbor interpolation (ANTs 2.2.0[78]) while keeping only voxels within the union of subject- and run-specific brain masks produced by the fMRIprep pipeline[55,56]. The resulting subject-specific ROIs therefore had varying number of voxels ($\mu = 768.14$, $\sigma = 65.62$, min = 667, max = 954).

## Verifying design trial-wise estimability.

To verify that the individual trials are estimable (for the trial-wise multivariate analysis) and as a control over multi-collinearity[94], we convolved a design matrix with the HRF for each subject with one regressor per stimuli (432 regressors with duration equal to the stimulus duration), two regressor across all cues (split by context) and three regressor for all outcomes (one for each EV). We then computed the VIF for each stimulus regressor (i.e., how predictive is each regressor by the other ones). None of the VIFs surpassed 1.57 across all trials and subjects ($\mu_{VIF} = 1.42$, $\sigma_{VIF} = 0.033$, min = 1.34). When repeating this analysis with a GLM in which also outcomes were split into trialwise regressors, we found no stimuli VIF larger than 3.09 ($\mu_{VIF} = 2.64$, $\sigma_{VIF} = 0.132$, min = 1.9). Note that 1 is the minimum (best) value and 5 is a relatively conservative threshold for collinearity issues (e.g. ref. 94). This means that the BOLD responses of individual trials can be modeled separately and should not have collinearity issues with other stimuli nor with the outcome presentation of each trial.

## Multivariate analysis

**RDM analyses.** RDM was conducted using betas taken from a GLM fit to data in native space (4 mm smoothing) with one onset for EV of 1D trials and one onset for each combination or EV and $EV_{back}$ for 2D trials (e.g., one onset for all trials where EV = 30 and $EV_{back}$ = 30, one onset when EV = 30 and $EV_{back}$ = 50, etc.). Duration of the onsets was set to 0. Regressors of no interest were identical to the GLMs described in "Univariate fMRI analysis" section above. For each subject, we extracted the beta values for each run from the above defined functional ROI for each one of the 2D onset regressors. We then performed multivariate noise normalization (normalize each voxel by its residuals[47]) and mean pattern subtraction (i.e., subtract the mean pattern across conditions for each voxel from each response pattern[47]). Lastly, we computed the Euclidean distance between each pair of patterns across runs using Nilearn[75]. Note that noise-normalized Euclidean distance is equivalent to the Mahalanobis distance[47]. To prevent biasing the diagonal, we excluded any correlation within a run across conditions (where the diagonal would be 1). This resulted in a 9 × 9 RDM for each subject and each block comparison. The resulting distances (half the matrix including the diagonal for each subject) were analyzed in R (R version 3.6.3[64], RStudio version 1.3.959[65]) with Generalized Linear Mixed Models using Template Model Builder (glmmTMB[95]) models, employing a gamma distribution family with a "inverse" link function. When describing main effects of models, the $\chi^2$ represents Type II Wald $\chi^2$ tests, whereas when describing model comparison, the $\chi^2$ represents the log-likelihood ratio test. Model comparison throughout the paper was done using the "anova" function. Throughout all the analyses, each regressor was scaled prior to fitting the models.

The best explaining model for the main effects of the RDM was:

$$d_{i,j}^k = \beta_0 + \gamma_{0k} + \beta_1 \text{Diagonal}_{EV} + \beta_2 \text{Diagonal}_{EV_{back}} + \zeta_{0,k,\text{frequency}} \quad (4)$$

where $d_{i,j}^k$ is the Mahalanobis distance of combination $i$ and $j$ for subject $k$, where $i$ and $j$ each represent all possible patterns (i.e., combination of EV and $EV_{back}$. $\beta_0$ and $\gamma_{0k}$ represent global and subject-specific intercepts. $\text{Diagonal}_{EV}$ is 1 when the EV of pattern $i$ is the same as the EV of pattern $j$. $\text{Diagonal}_{EV_{back}}$ is 1 when the $EV_{back}$ of pattern $i$ is the same as $EV_{back}$ of pattern $j$. $\zeta_{0,k,\text{frequency}}$ is an additional intercept for every level of frequency nested within each within each subject level. For details on the effect of frequency, see Fig. S6.

The best explaining model for the value difference effects of the RDM was:

$$d_{i,j}^k = \beta_0 + \gamma_{0k} + \beta_1 \text{ValueDifference}_{EV} + \beta_2 \text{ValueDifference}_{EV_{back}} + \zeta_{0,k,\text{frequency}} \quad (5)$$

where all parameters are identical to Eq. (4) above, only that $\text{ValueDifference}_{EV}$ corresponds to the value difference between the EV of pattern $i$ and the EV of pattern $j$ and $\text{ValueDifference}_{EV_{back}}$ is the value difference between the $EV_{back}$ of pattern $i$ and the $EV_{back}$ of pattern $j$.

**Classification procedure.** The training set for Value and Context classifiers consisted of fMRI data from behaviorally accurate 1D trials. For each trial, we took the TR corresponding to ~5 s after stimulus onset (round(onset + 5)) to match the peak of the Haemodynamic Response Function (HRF) estimated by SPM[90]. Training of Value and Context classifiers was done using a leave-one-run-out scheme across the four runs with 1D trials. To avoid bias in the training set after sub-setting only to behaviorally accurate trials (i.e., over-representation of some information) we up-sampled each training set to ensure equal number of examples in the training set for each combination of EV (3), Context (2) and Chosen-Side (2). Specifically, if one particular category was less frequent than another (e.g., more value-30, left, color trials than value-50, left-color trials) we up-sampled that example category by randomly selecting a trial from the same category to duplicate in the training set, whilst prioritizing block-wise balance (i.e., if one block had 2 trials in the chunk and another block had only 1, we first duplicated the trial from under-represented block, etc.). We did not up-sample the testing set. The $EV_{back}$ classifiers were trained on behaviorally accurate 2D trials (5 s after stimulus onset) and up-sampled by EV (3), Context (2) and $EV_{back}$ (3) (without Chosen-Side as this resulted in excluding many subjects for lack of trials in some training sets). Due to strong imbalance of unique examples of $EV_{back}$ in the training sets (see below) we trained 3 one-vs.-rest classifiers, each tasked with identifying one level of $EV_{back}$. This required to adjust the sample weights in order to account for the higher frequency of the "rest" compared to the "one" label.

Decoding was conducted using multinomial logistic regression as implemented in scikit-learn 0.22.2[96], using a $C$ parameter of 1.0, L2 regularization and the lbgfs solver. For each test example (i.e., trial) we obtained the predicted probability per class. To avoid numerical issues in the subsequent modeling of the classifier's predictions, probabilities were constrained to lie within 0.00001 and 0.99999, rather than 0 and 1. In addition to the probabilities, we obtained the balanced classification accuracy (i.e., is the class with the highest probability also the correct class of the test trial). We separately averaged classification for each participant, test fold and label (this ensured controlling for any label imbalance in the testing set).

In the classification analyses we modeled directly the class probabilities estimated by the classifiers with beta regression mixed effects models[48]. For technical reasons, before modeling the probabilities using linear mixed effects models, we averaged the classifiers probabilities across the nuisance effects, i.e., we obtained one average probability for each combination of relevant and irrelevant values. Crossing each level of EV (three levels) with each level of irrelevant value of the chosen side combined with irrelevant value of the non-chosen side (12 level, see Fig. 1), resulted in 36 combinations per participant. Note that the relevant value of the unchosen cloud was always EV - 20 and therefore we did not include this as a parameter of interest. After averaging, we computed for each combination of values the $EV_{back}$, Congruency and alternative parameters (see Fig. S9). The main model comparison, as well as the lack of effects of any nuisance regressor, was confirmed on a dataset with raw, i.e., non-averaged, probabilities (see Figs. S7 and S9). Because in the one-vs.-rest training of $EV_{back}$ classifiers the three class probabilities for each trial were obtained independently, they sum to 1. We therefore first normalized the probabilities for each testing trial.

Probabilities were analyzed in R (R version 3.6.3[64], RStudio version 1.3.959[65]) with Generalized Linear Mixed Models using Template Model Builder (glmmTMB[95]) models, employing a beta distribution family with a "logit" link function. When describing main effects of models, the $\chi^2$ represents Type II Wald $\chi^2$ tests, whereas when describing model comparison, the $\chi^2$ represents the log-likelihood ratio test. Model comparison throughout the paper was done using the "anova" function. Throughout all the analyses, each regressor was scaled prior to fitting the models. Lastly, for the analysis of behavioral accuracy (Fig. 6) we also included behaviorally wrong trials.

Additional coding of the analyses in Python (3.7[97]) using NumPy (1.19.5[98]) and pandas (1.1.5[99]). Most of the plots were produced using ggplot2 (3.3.5[100]).

**Value similarity analyses.** Asked whether the predicted probabilities reflected the difference from the objective probability class.

The model we found to best explain the data was:

$$P_{t,c}^k = \beta_0 + \gamma_{0k} + \beta_1|EV_t - c_t| + \beta_2|EV_t - c_t|EV_{back_t} \qquad (6)$$

where $P_{t,c}^k$ is the probability that the Value classifier assigned to class $c$ in trial $t$ for subject $k$, $\beta_0$ and $\gamma_{0k}$ represent global and subject-specific intercepts, $|EV_t - Class_{c,t}|$ is the absolute difference between the EV of the trial and the class the probability is assigned to and $|EV_t - Class_{c,t}|EV_{back_t}$ is the interaction of this absolute difference with $EV_{back}$. For models nested in the levels of EV, we included $\zeta_{0_{k,EV}}$, which is the EV-specific intercept nested within each within each subject level. In these models, testing for main effects of $EV_{back}$ or Congruency was not sensible because both factors do not discriminate between the classes, but rather assign the same value to all three probabilities from that trial (which sum to 1). More details can be found in Fig. S7.

**Values, not perceptual features and not attention capture, explain our effects best.** For the feature similarity model we substituted $|EV_t - c_t|$ from Eq. (6) with a "similarity" parameter that encoded the perceptual similarity between each trial in the test set and the perceptual features that constituted the training examples of each class of the classifier. For 1D trials, this perceptual parameter was identical to the value similarity parameter ($|EV_t - c_t|$). This was because from the shown pairs of colors, both colors overlapped between training and test if the values were identical; one color overlapped if the values were different by one reward level (e.g., a 30 vs. 50 comparison corresponded to two trials that involved pink vs. green and green vs. orange, i.e., sharing the color green); and no colors overlapped if the values were different by two levels (30 vs. 70). On 2D trials however, due to changing background features and their value-difference variation, perceptual similarity of training and test was not identical to value similarity. Even though both the value similarity and the perceptual similarity parameter correlated ($\rho = 0.789$, $\sigma = 0.005$), we found that the value similarity model provided a better AIC score (value similarity AIC: −3898, Feature similarity AIC: −3893, Fig. S7d). Detailed description with examples can be found in Fig. S7. Crucially, even when keeping the value difference of the irrelevant features at 20, thus limiting the testing set only to trials with feature-pairs that were included in the training, our value similarity model provided a better AIC (−1959) than the feature similarity model (−1956). To test for a perceptual alternative of $EV_{back}$ we substituted the corresponding parameter from the model with $Similarity_{back}$. This perceptual parameter takes on 1 if the perceptual feature corresponding to the $EV_{back}$ appeared in the 1D training class (as highest or lowest value) and 0 otherwise. As described in the main text, none of the perceptual-similarity encoding alternatives provided a better fit than our models that focused on the expected values the features represented.

**Modeling the influence of irrelevant values and Context signals on EV representation.** The following model of the probability of the objective EV was found to explain the data best:

$$P_{t,EV}^k = \beta_0 + \gamma_{0k} + \beta_1 EV_{back_t} + \beta_2 P_{t,Context}^k \qquad (7)$$

where $P_{t,EV}^k$ is the probability assigned to the objective class by the Value classifier (corresponding to EV of the trial $t$) for subject $k$, $\beta_0$ and $\gamma_{0k}$ represent global and subject-specific intercepts, $EV_{back}$ is the maximum of the two ignored values (or the EV of the contextually irrelevant context) and $P_{t,Context}^k$ is the probability assigned to the

objective class by the Context classifier (logit-transformed, i.e., $\text{logit}(P) = \log \frac{P}{1-P}$, and scaled for each subject). For models nested in the levels of EV, we included $\zeta_{0_{k,\text{EV}}}$ which is EV specific intercept nested within each within each subject level (see Fig. S9). Investigations of alternative parametrizations of the values can be found in Fig. S9. Including an additional regressor that encoded trials in which EV = EV$_{\text{back}}$ (or: match) which did not improve model fit, and no evidence for an interaction of the match regressor with the EV$_{\text{back}}$ was found (LR test with added terms: $\chi^2_{(1)} = 0.45$, $p = 0.502$, $\chi^2_{(1)} = 0.77$, $p = 0.379$, respectively). This might indicate that when value expectations of both contexts matched, there was neither an increase nor a decrease of $P_{\text{EV}}$.

To compute the correlations between each pair of classes we transformed the probabilities for each class using a multinomial logit transform. For example, for class 30 we performed probabilities were transformed with $\text{mlogit}(P_{t,30}) = 0.5(\log \frac{P_{t,30}}{P_{t,50}} + \log \frac{P_{t,30}}{P_{t,70}})$. To examine the relationship between EV and EV$_{\text{back}}$, we only included 2D trials in which EV $\neq$ EV$_{\text{back}}$. This allowed us to categorize all three probabilities as either EV, EV$_{\text{back}}$ or Other, whereby Other reflected the value that was neither the EV, nor the EV$_{\text{back}}$. To prevent bias we included only trials in which Other was presented on screen (as relevant or irrelevant value). We then averaged across nuisance regressors (see "Classification procedure") and computed the correlation across all trials (Spearman rank correlation). Lastly, we Fisher z-transformed the correlations ($0.5 \log \frac{1+\rho}{1-\rho}$) to approximate normality for the $t$ test. To validate these results, we performed an additional model comparison in which we added a term of the logit transformed $P_{\text{EV}_{\text{back}}}$ or of $P_{\text{other}}$ to Eq. (7) ($\beta_2 \text{mlogit}(P_{t,\text{EV}_{\text{back}}})$ or $\beta_2 \text{mlogit}(P_{t,\text{Other}})$, respectively). As reported in the main text, adding a term reflecting $P_{\text{EV}_{\text{back}}}$ resulted in a smaller (better) AIC score than when we added a term for $P_{\text{other}}$ ($-567$, $-475$, respectively). This was also preserved when running the analysis including nuisance regressors (see vs. in Eq. (2)) on the non-averaged data (AICs: $-5913.3$, $-5813.3$). We note that subsetting the data the way we did resulted in a strong negative correlation in the design matrix between EV and EV$_{\text{back}}$ ($\rho = -0.798$, averaged across subjects). Although this should not directly influence our interpretation, we validated the results by using alternative models with effects hierarchically nested within the levels of EV and EV$_{\text{back}}$ (Averaged data AICs: $-560$, $-463$, Raw data AICs: $-5906.8$, $-5804.3$).

As previously clarified, $P_{\text{EV}_{\text{back}}}^{2D}$ was derived from a classifier trained on 2D trials. The number of unique examples for each class of EV$_{\text{back}}$ differed drastically (due to our design, see Figs. 1c and S6), which motivated us to split the decoding of EV$_{\text{back}}$ to three classifiers, each trained on a different label (see "Classification procedure"). However, our approach of combining one-vs.-rest training with oversampling and sample weights could not fully counteract these imbalances and a balanced accuracy did not surpass chance level ($t$-test against chance: $t_{(34)} = 0.96$, $p = 0.171$) and the probabilities each classifier assigned to its corresponding class ($P_{\text{EV}_{\text{back}}}^{2D}$) were still biased by class imbalances. Specifically, the correlation of $P_{\text{EV}_{\text{back}}}^{2D}$ and EV$_{\text{back}}$ was $\rho_\mu = 0.26$, $\rho_\sigma = 0.07$ across subjects, where "2D" indicates the classifier was directly trained on 2D trials, unlike with $P_{\text{EV}_{\text{back}}}$ which comes from a classifier trained on EV in 1D trials. Since in this analysis we were mainly interested in the neural representation of EV$_{\text{back}}$ regardless of whether EV$_{\text{back}}$ was 30, 50 or 70 in given trial, we solved this issue by using mixed effect models and setting a random intercept for each level of EV$_{\text{back}}$ (i.e., running the models nested within the levels of EV$_{\text{back}}$). Importantly, due to the symmetric nature of the RDM, this trial frequency bias is orthogonal to the main effect of EV$_{\text{back}}$ reported earlier (Fig. S6a–c).

Thus, when testing across the levels of EV$_{\text{back}}$, the model that best explained the data was:

$$P_{t,\text{EV}}^k = \beta_0 + \gamma_{0k} + \beta_1 \text{EV}_{\text{back}_t} + \beta_2 P_{t,\text{Context}}^k + \beta_3 P_{t,\text{EV}_{\text{back}}}^{k,2D} + \beta_4 P_{t,\text{Context}}^k P_{t,\text{EV}_{\text{back}}}^{k,2D} + \zeta_{0_{k,\text{EV}_{\text{back}}}}$$
$$(8)$$

where similar to Eq. (7), $P_{t,\text{EV}}^k$ is the probability assigned to the EV class by the Value classifier for trial $t$ and subject $k$, $\beta_0$ and $\gamma_{0k}$ represent global and subject-specific intercepts and $P_{t,\text{Context}}^k$ is the logit-transformed probability assigned to Context class. $P_{t,\text{EV}_{\text{back}}}^{k,2D}$ is the probability the EV$_{\text{back}}$ classifier assigned the correct class (in main text: $P_{\text{EV}_{\text{back}}}^{2D}$, where 2D notes that this classifier was trained on 2D trials) and $\zeta_{0_{k,\text{EV}_{\text{back}}}}$ is EV$_{\text{back}}$ specific intercept nested within each within each subject level.

**Linking MRI effects to behavior.** When modeling the probability of EV$_{\text{back}}$ from the Value classifier ($P_{\text{EV}_{\text{back}}}$, Fig. 6a), we did not average across nuisance regressors. Our baseline model was: $P_{t,\text{EV}_{\text{back}}}^k = \beta_0 + \gamma_{0k} + \nu_1 \text{side}(t) + \nu_2 \text{switch}(t) + \nu_3 \text{Context}(t)$. Neither including a main effect nor interactions between EV, EV$_{\text{back}}$ and Congruency improved model fit. When including behaviorally wrong trials in the model, we used `drop1` in combination with $\chi^2$-tests from `lmer4` package[66] to test which of the main effects or interactions improves the fit. This resulted in the following model as best explaining the data:

$$P_{t,\text{EV}_{\text{back}}}^k = \beta_0 + \gamma_{0k} + \beta_1 \text{EV}_t \times \text{EV}_{\text{back}_t} + \beta_2 \text{Congruency}_t \text{Accuracy}_t$$
$$+ \nu_1 t + \nu_2 \text{side}_t + \nu_3 \text{switch}_t + \nu_4 \text{Context}_t \qquad (9)$$

where $P_{t,\text{EV}_{\text{back}}}^k$ is the probability the Value classifier assigned to the EV$_{\text{back}}$ class (corresponding to EV$_{\text{back}}$ of trial $t$) for subject $k$, $\beta_0$ and $\gamma_{0k}$ represent global and subject-specific intercepts, EV is the maximum of the two relevant and EV$_{\text{back}}$ is the maximum of the two ignored values. Congruency reflects whether the actions chosen in the relevant vs. irrelevant context would be the same, and the Accuracy regressor has 1 if participants chose the highest relevant value and 0 otherwise. We note that the interaction EV $\times$ EV$_{\text{back}}$ ($\chi^2_{(1)} = 4.18$, $p = 0.041$) indicates higher in trials in which EV and EV$_{\text{back}}$ were more similar, the probability assigned to EV$_{\text{back}}$ was higher. However, we find this effect hard to interpret since this corresponds to the value similarity effect we previously reported.

In order to investigate the effect of vmPFC neural representations on behavioral accuracy, we used hierarchical model comparison to directly test the influence of neural representation of EV, EV$_{\text{back}}$ and Context on behavioral accuracy separately for congruent and incongruent trials (Fig. 6b, c). First, we tested if adding $\text{logit}(P_{t,\text{Context}})$, $\text{mlogit}(P_{t,\text{EV}})$ or $\text{mlogit}(P_{t,\text{EV}_{\text{back}}})$ to Eq. (3), would help to explain the behavioral accuracy better. Because the analysis was split for congruent and incongruent trials, we excluded the terms involving a Congruency effect. For incongruent trials, only $\text{logit}(P_{t,\text{Context}})$ improved the fit (LR-tests: $\text{logit}(P_{t,\text{Context}})$: $\chi^2_{(1)} = 3.66$, $p = 0.055$, $\text{mlogit}(P_{t,\text{EV}})$: $\chi^2_{(1)} = 0.28$, $p = 0.599$, $\text{mlogit}(P_{t,\text{EV}_{\text{back}}})$: $\chi^2_{(1)} = 0.0$, $p = 0.957$). In a second step we then separately tested the interactions $\text{logit}(P_{t,\text{Context}}) \times \text{mlogit}(P_{t,\text{EV}})$ or $\text{logit}(P_{t,\text{Context}}) \times \text{mlogit}(P_{t,\text{EV}_{\text{back}}})$ and found that only the latter had improved the fit ($\chi^2_{(1)} = 1.78$, $p = 0.183$, $\chi^2_{(1)} = 6.33$, $p = 0.012$, respectively). For congruent trials, only $\text{mlogit}(P_{t,\text{EV}_{\text{back}}})$ and marginally $\text{mlogit}(P_{t,\text{EV}})$ improved the fit (LR-tests: $\text{logit}(P_{t,\text{Context}})$: $\chi^2_{(1)} = 0.0$, $p = 0.922$, $\text{mlogit}(P_{t,\text{EV}})$: $\chi^2_{(1)} = 3.5$, $p = 0.061$, $\text{mlogit}(P_{t,\text{EV}_{\text{back}}})$: $\chi^2_{(1)} = 6.48$, $p = 0.011$). In a second step we tested separately the interactions $\text{logit}(P_{t,\text{Context}}) \times \text{mlogit}(P_{t,\text{EV}})$, $\text{logit}(P_{t,\text{Context}}) \times \text{mlogit}(P_{t,\text{EV}_{\text{back}}})$ or $\text{mlogit}(P_{t,\text{EV}_{\text{back}}}) \times \text{mlogit}(P_{t,\text{EV}})$ and found none of these improved model fit when adding them to a model that included both main effects from the previous step ($\chi^2_{(1)} = 0.34$, $p = 0.560$, $\chi^2_{(1)} = 0.278$, $p = 0.598$, $\chi^2_{(1)} = 2.49$, $p = 0.115$, respectively).

To investigate the effect of vmPFC neural representations on RT in behaviorally accurate trials, we asked whether subjects who had a stronger effect of Context representation ($P_{\text{context}}$) on EV representation ($P_{\text{EV}}$) or a stronger Spearman rank correlation between $P_{\text{EV}}$ and $P_{\text{EV}_{\text{back}}}$ (taken from the Value classifier) also had a stronger effect of Congruency on their RT. Additionally, we asked whether subjects who had a stronger effect of EV$_{\text{back}}$ on $P_{\text{EV}}$ and or a stronger effect of $P_{\text{EV}_{\text{back}}}^{k,2D}$ on $P_{\text{EV}}$ also had a stronger modulation of EV$_{\text{back}}$ on the Congruency RT

effect. To obtain subject specific effect of Congruency on RT we added $\gamma_{1k}$ Congruency and $\gamma_{2k}$ CongruencyEV$_{\text{back}_t}$ to the RT model (Eq. (2)), representing subject-specific slopes of Congruency for subject $k$ and for the interaction of Congruency and EV$_{\text{back}}$, respectively. The subject-specific correlation of $P_{\text{EV}}$ and $P_{\text{EV}_{\text{back}}}$ was estimated by using only trials in which EV $\neq$ EV$_{\text{back}}$. Probabilities were multinomial logit transformed and correlations were Fisher $z$-transformed (see above) before averaging across trials to achieve one correlation value per subject. In the main text and in Fig 5e, f we did not average the data to achieve maximum sensitivity to trial-wise variations. The results reported in the main text replicate when running the same procedure while averaging the data across nuisance regressors following the multinomial logit transformation ($R = 0.38$, $p = 0.023$). To extract subject-specific slopes for the effect of EV$_{\text{back}}$ on $P_{\text{EV}}$ we included a term for this effect ($\gamma_{1k}$EV$_{\text{back}_t}$) in Eq. (7), but due to convergence issues during model fitting, we had to drop the subject-specific intercept ($\gamma_{0k}$) in that model. Similarly, to extract subject-specific slopes for the effect of $P^{2D}_{\text{EV}_{\text{back}}}$ on $P_{\text{EV}}$ we included a term for this effect ($\gamma_{1k}P^{k,2D}_{t,\text{EV}_{\text{back}}}$) in Eq. (8).

### Reporting summary

Further information on research design is available in the Nature Portfolio Reporting Summary linked to this article.

## Data availability

Behavioral data can be found in https://git.mpib-berlin.mpg.de/moneta/parallelrepresentation. All individual fMRI datasets can be found at https://gin.g-node.org/nirmoneta/SODIVA and are shared under Creative Commons Attribution-ShareAlike 4.0 International Public License (see LICENSE file in repository). We supply the fMRI data needed to reproduce the findings presented in the manuscript, i.e., conventionally preprocessed data (fmriprep[71,72]) from the functionally defined vmPFC ROI (smoothed at 4 and 8 mm, in MNI and native space). We additionally share data from various steps of the analyses: defaced T1 images, functionally defined ROIs in MNI and individual native space, preprocessed data ready to be classified including individual classifier decoding results, individual RSAs (see README in https://git.mpib-berlin.mpg.de/moneta/parallelrepresentation for full details on the data folder structure). In case of interest in the whole brain raw data, please contact the corresponding authors. Source data are provided with this paper.

## Code availability

Custom code for the task, behavioral analyses, preprocessing of fMRI data as well as fMRI analyses to reproduce the findings presented in the manuscript have been deposited in https://git.mpib-berlin.mpg.de/moneta/parallelrepresentation under Creative Commons Attribution-ShareAlike 4.0 International Public License (see LICENSE file in repository).

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

## Acknowledgements

N.W.S. was funded by an Independent Max Planck Research Group grant awarded by the Max Planck Society (M.TN.A.BILD0004) and a Starting Grant from the European Union (ERC-2019-StG REPLAY-852669) and the Federal Ministry of Education and Research (BMBF) and the Free and Hanseatic City of Hamburg under the Excellence Strategy of the Federal Government and the Länder. N.M. was funded by and is grateful for a scholarship from the Ernst Ludwig Ehrlich Studienwerk (ELES) and Einstein Center for Neuroscience (ECN) Berlin throughout this study. We thank Angela J. Langdon for comments on the manuscript. We thank Gregor Caregnato for help with participant recruitment, Anika Löwe, Lena Maria Krippner, Sonali Beckmann and Nadine Taube for help with data acquisition, all participants for their participation and the Neurocode lab for numerous contributions and help throughout this project.

## Author contributions

The following list of author contributions is based on the CRediT taxonomy (for details: refs. 101). N.M. and N.W.S. contributed to conceptualization, formal analysis, funding acquisition, methodology, project administration, software, validation, visualization and writing the original draft, reviewing and editing. N.W.S. supervised the project and provided the resources. N.M. contributed to data curating and investigation. M.M.G. and H.R.H. consulted at numerous steps of the planning, analysis and writing of the manuscript.

## Funding

## Competing interests

M.M.G. is employee of Aya Technologies Ltd. The rest of the authors declare no competing interests.
