## [Peer Review File · Nature Communications]

Task state representations in vmPFC mediate relevant and irrelevant value signals and their behavioral influenceREVIEWER COMMENTS

Reviewer #1 (Remarks to the Author):

It has been a pleasure to read the manuscript from Moneta and colleagues titled "Representation of values in vmPFC compete for guiding behaviour". The authors investigated (using multivariate fMRI methods) the representational contentment of vmPFC to relevant and irrelevant dimensions. Participants performed a perceptual discrimination task, characterised by 2 dimensions (motion and color). During the task the experimenters were communicating to the participants the relevant dimension (i.e., the rewarded dimension) valid for a short but variable number of trials (2 – 7). This generated two distinct contexts that were explicitly communicated to the participants. The authors then tested how the irrelevant condition affected behavior. The main behavioural findings were that the participants responded faster for choice that elicited higher rewards (in line with previous studies) and that, critically, the irrelevant context slowed the response (directly proportionally to the size of the irrelevant reward). They then investigated how the irrelevant dimension affected the neural representation in vmPFC during the relevant choice. They found that the irrelevant value affected the representation in vmPFC. They did not find direct (statistically significant) evidence of the representation of irrelevant background values using a classifier trained on each EV class probably due to a lack of power (probably worth to train the classifier on 2 classes, high and low value, using a median split to increase power). However, they find other (more convincing) evidence that irrelevant value interfered (dampened) the representation in vmPFC of the relevant value and linked these results to slowness in RT and choice error. They concluded the set of analysis confirming that the neural signature for irrelevant values was not detectable using a classic univariate approach.

The paper is overall solid, the task has been carefully crafted and the analyses presented are robust and thorough. I find most of the results convincing. I have, therefore, little to advise on this side. The presentation of the results might be streamlined a bit. Sometimes, it is hard to distinguish the key analysis for the corollary ones making the reading of the manuscript sometime fatiguing.

My only real concern is about the general conclusions that one can derive from this (well executed) study. From the title, abstract (but also introduction and discussion) one is under the impression that this study speaks to general problem of how humans engage and negotiate amongst different value representations. How the same object can lead to very different outcomes depending on the context. How these multiple representations interplay shaping behaviour and neural representations. I believe that these are indeed very important questions for which we still don't have convincing answers. However, I am slightly doubtful that this work contributes to address any of these questions. My main problem is that the task used here speaks more directly to a problem of cognitive control than value-based choice process. The field of cognitive control has studied for many years how distinct demands/context can produce interference in behaviour. Differences that are reflected in the underpinning neural representations. One does not need to go far to find examples, such the extremely well known stroop task, in which irrelevant/incongruent dimension impact behaviour in a way that looks very similar to the effects presented here. In defence, one might argue that what makes this a value-based task is that to different combination of stimuli dimensions was associated different number of points, leading to variable amount of reward, and a feature that the author showed that modulated behaviour. However, this is not dissimilar to manipulating the saliency (or reward) of some perceptual dimensions in a classic cognitive control setting. There are many studies (not discussed here) in which this has been done more or less explicitly. Another important feature of this task, that further highlighting the cognitive control aspect, is that the two contexts alternated very often during the task. This might have exacerbated the interference observed, making it a by-product of the specific task setup rather than a general feature of value computation. I am indeed curious if the effect of interference was modulated by the lags between the context switch (although given the small range might not be the case). I would like to hear what the authors think and hopefully see some further analyses that might mitigate my worries. In any case, I strongly advise the authors to engage more with cognitive control literature and discuss more in detail these issue and potential caveats, including the problem of rapid switches between contexts. Currently most of the

discussion is a recapitulation and summary of the results with very little actual discussion. I believe that the summary can be moved at the end of each result section (improving also clarity – see other point) leaving plenty of space for a more extensive discussion.

My main worry is that without more empirical evidence and extensive discussion about this issue the interpretation of these results might result misleading for the field of value-based decisions (I am sure that this is not the author intended).

I hope that the authors will find my feedbacks useful.

Reviewer #3 (Remarks to the Author):

Moneta et al. examine how context and value are represented in vmPFC during a contextual value-based decision making task. They show that multivariate patterns in vmPFC contain information about context and contextual value. Furthermore, they leverage behavioral impacts of context-irrelevant values and trial-by-trial analyses of decoder accuracy to examine how context-irrelevant value information can interfere with stored values when context representations are weak.

The paper is interesting, timely, and the research is generally well designed and described. It uses neuroimaging analyses innovatively to tackle mechanistic questions that are typically off limits for human studies, and I think that it makes great strides in doing so. That said, I have a number of concerns with the paper in its current form, as well as questions about what exactly was done, and why. A complete listing of my comments is below.

Major concerns:

1) I was a bit surprised to see that the authors do not perform deconvolution to attain single trial activations for the multivariate analyses, but instead use a single TR at the peak of the HRF. This approach seems like it could have at least two potential shortcomings. One is that timing variability could be mis-attributed to amplitude variability. It appears the authors do not have jitter in their task timings (though it would be useful to state this explicitly). Even so, the participant responses are variable, making it impossible to get around this issue completely. The second shortcoming is that it leaves potentially informative data on the table – that is to say that TRs before and after the peak may provide some additional information about the activation that is not picked up using the current methods. The degree to which this is true is complicated, as it depends on the temporal autocorrelation of both signal and noise, as well as task design, but at minimum it would be useful for the authors to provide some justification for their choice, and it would be even more compelling if they could demonstrate that the key effects hold when using the deconvolution approach.

2) Some key reported results rely on the degree to which patterns of activation in vmPFC resemble those most diagnostic of high EV_back trials. Yet, there is very little evidence to suggest that vmPFC represents EV_back. Decoding of EV_back category was not significantly above chance, and thus interpreting the key results that rely on EV_back classifiers seems a bit tricky. Is EV_back represented in vmPFC, as the authors suggest... or does training a classifier on EV_back for some reason pick up on noise dimensions that have some interesting relationship to EV, and therefore behavior? The original design was balanced, but the need to oversample specific trials makes this a distinct possibility, and thus a bit hard to tell. Could the authors demonstrate successful classification of EV_back in specific individuals? Or perhaps do analyses that would allow them to understand why EV_back might not be decodable, even if the information were present? I could imagine other methods for detecting multivariate representations, such as RSA, could be helpful in this regard. As it stands, I think that there is a lot of ambiguity on how to interpret results that rely on the EV_back classifier, and if additional support for the representation of EV_back in vmPFC is not provided, then specific mechanistic claims would need to be substantially scaled back.

3) I was a bit confused as to why the authors use categorical classification procedures throughout. The authors seem to think the vmPFC is representing a continuous value signal – so why not predict continuous value from multivariate activation, rather than the category? Doing so would directly assess the dimensions over which appropriate context and inappropriate context values are represented (ie. the weights from the predictive model). In this case the authors could just examine the cosine angle between these dimensions to see whether representation of the two value signals (context appropriate/context inappropriate) is orthogonal in those participants who show less interference, and perhaps non-orthogonal in those participants who show more.

4) Even if the authors do have a good reason not to use a continuous decoding strategy – it still seems like it would be useful to understand the relationship between the multivariate codes that they measure. For example, in existing models, do the readout weights for the highest EV look similar to those for the highest EV_back? Or are they completely independent?

5) Throughout manuscript, there were p values reported without adequate description of the statistical test and null hypothesis. For example, in figure 1a, but also throughout main text. In some cases, statistical tests were implied (Chi squared values reported, and presumably likelihood ratio tests were performed) but I was a bit unsure why – since in some cases both variables examined were continuous and models nested, such that I expected to see a nested-F test. Clarification and adequate explanation in main text would be helpful.

6) In general, AIC values without any associated uncertainty measures or confusion matrices to validate them are a bit tricky to interpret. I also found likelihood ratio test statistics a bit tough to interpret – because in many cases the direction of the coefficients assigned to the coefficient of interest was more critical to the overarching hypothesis than the amount of variance explained. In most cases I think that relevant parameter estimates, along with their 95% confidence intervals, would be a more useful way to display results from the statistical modeling.

Minor concerns:

I was a bit confused by the logic relating to the predicted interaction between context, EV_back, and EV (lines 348-9). Is the idea that you expect to see EV_back representations to have minimal contribution to the accuracy of EV representations when context representations are confident and accurate, but more influence when they are inaccurate? This was originally my interpretation, but this is not clear from the lines referenced above, and I think a schematic illustration of the author's prediction along with a display of the results in the same format could be helpful here.

In Figure 1e it would be useful to clarify in figure that the matrix displayed is representing the context-irrelevant feature pairs. This info is in caption, but could be a bit more clearly communicated.

Throughout paper, there are plots that include $\logit(\text{prob})$ on the x-axis in which model predictions are displayed, however it is unclear the range of that axis that actually contains data. It would be useful if some markers for the range of the actual data could be provided, so the magnitude of the effects easily interpreted.

I think that there are a couple of references to figure 5 that should actually be figure 4.

Line 345: independent is probably not the right word here.

The terminology used to describe trial-to-trial variability in classifier reports could use improvement. Terms like "strongly decoded" seem misleading, given overall decoding accuracy, and that what they are really trying to say is that this trial has a high amount of some multivariate feature.

In figure 6a, decoding of pEV_back for congruent trials is at (or below) chance – but presumably

this is only because the classifier was trained on all trial types, right?

Figure 6b: legend would be helpful in this figure.

The authors summary states that accuracy and RT data support their claims – but I do not see results from choice regression in main paper, unless I have missed something.

Point by point replies

Reviewer 1

Reviewer 1, comment 1:

It has been a pleasure to read the manuscript from Moneta and colleagues titled “Representation of values in vmPFC compete for guiding behaviour”. The authors investigated (using multivariate fMRI methods) the representational contentment of vmPFC to relevant and irrelevant dimensions. Participants performed a perceptual discrimination task, characterised by 2 dimensions (motion and color). During the task the experimenters were communicating to the participants the relevant dimension (i.e., the rewarded dimension) valid for a short but variable number of trials (2 – 7). This generated two distinct contexts that were explicitly communicated to the participants. The authors then tested how the irrelevant condition affected behavior. The main behavioural findings were that the participants responded faster for choice that elicited higher rewards (in line with previous studies) and that, critically, the irrelevant context slowed the response (directly proportionally to the size of the irrelevant reward). They then investigated how the irrelevant dimension affected the neural representation in vmPFC during the relevant choice. They found that the irrelevant value affected the representation in vmPFC.

They did not find direct (statistically significant) evidence of the representation of irrelevant background values using a classifier trained on each EV class probably due to a lack of power (probably worth to train the classifier on 2 classes, high and low value, using a median split to increase power). However, they find other (more convincing) evidence that irrelevant value interfered (dampened) the representation in vmPFC of the relevant value and linked these results to slowness in RT and choice error. They concluded the set of analysis confirming that the neural signature for irrelevant values was not detectable using a classic univariate approach.

Response:

Thank you for the positive feedback about our manuscript.

The lack of direct evidence for irrelevant background values was also brought up by another reviewer (Rev. 3, Comment 2), and has now been extensively addressed.

Specifically, we came to the conclusion that the class imbalance (different frequency of each EV/back level, see Fig 3 below) represented the major obstacle for the successful decoding of irrelevant values, and therefore ran RSA analyses using betas estimated from HRF convolved regressors in a GLM (the GLM approach was used to address a major concern from Reviewer 3). RSA Analyses are known to be sensitive to multivariate effects and the GLM step remedies the imbalance problem (to some extent). In short, we fitted a GLM with one separate regressor for each combination of EV (the expected value) and EVBACK (the irrelevant value), irrespective of the context (cross-validated, 1D trials modeled separately). After multivariate noise normalization and mean pattern subtraction (Walther et al. 2016), we computed the Mahalanobis distance of each regressor combination resulting in a 9 by 9 Representational Dissimilarity Matrix (RDM, see Figure below and manuscript) for each subject. We then fitted mixed effects models to test whether pattern dissimilarities are sensitive to EV and EVback. Indeed, as expected, adding a main effect for EV dissimilarity (0 when two regressors share the same EV, 1 otherwise) improved model fit compared to a null model (LR-test: $\chi^2(1)=10.89$, $p<.001$, panel a). Strikingly, adding a main effect of EVBACK dissimilarity (0 when sharing EVBACK and 1 otherwise) further improved model fit (LR-test with added term: $\chi^2(1)=247.67$, $p<.001$, panel b). We believe this provides direct evidence of EVBACK representations in vmPFC and would like to thank the reviewer for making this helpful suggestion.

Crucially, because of the symmetric nature of the RSA, in this analysis the effect of the frequency of EVBACK classes (which biased the classifier) is orthogonal to the main effect of EVBACK and its value similarity - as can be seen in panels c and d of the new Figure 3 in the manuscript. We report now on page 8 of the revised manuscript:

“We ... tested whether both relevant and irrelevant expected values are reflected in multivariate vmPFC patterns using RSA. To estimate value-related activity patterns within the vmPFC mask, we fitted a General Linear Model (GLM) with one separate regressor for each combination of EV and EVback, irrespective of the context (cross-validated, 1D trials modeled separately) ... This resulted in one 9 × 9 Representational Dissimilarity Matrix (RDM, Fig.3 and online methods) per subject ... adding a main effect for EV dissimilarity (0 when two regressors share the same EV, 1 otherwise) improved model fit compared to a null model (LR-test: $\chi^2(1) 197 = 10.89, p < .001, \text{Fig.3b}$) ... Strikingly, adding a main effect of EVback dissimilarity (0 when sharing EVback and 1 otherwise) further improved model fit (LR-test with added term: $\chi^2(1) 200 = 247.67, p < .001, \text{Fig.3c}$).“

Reviewer 1, comment 2:

The paper is overall solid, the task has been carefully crafted and the analyses presented are robust and thorough. I find most of the results convincing. I have, therefore, little to advise on this side. The presentation of the results might be streamlined a bit. Sometimes, it is hard to distinguish the key analysis for the corollary ones making the reading of the manuscript sometime fatiguing.

Response:

Thank you for the comment, we are excited to hear that our results are overall convincing. We agree that the paper is dense and we did our best to streamline the results even more. The results section is now structured into paragraphs which are titled to summarize the following main finding. The paragraphs are also finished by a short summary section. We also shortened the results section overall. One main change is that we moved the section on Value similarity analysis to the SI, and replaced it by an RSA analysis (see above) that is easier to follow and also strengthens our evidence overall. We believe that together with a number of stylistic changes throughout the paper, the manuscript is much improved.

Reviewer 1, comment 3:

My only real concern is about the general conclusions that one can derive from this (well executed) study. From the title, abstract (but also introduction and discussion) one is under the impression that this study speaks to general problem of how humans engage and negotiate amongst different value representations. How the same object can lead to very different outcomes depending on the context. How these multiple representations interplay shaping behaviour and neural representations. I believe that these are indeed very important questions for which we still don't have convincing answers. However, I am slightly doubtful that this work contributes to address any of these questions. My main problem is that the task used here speaks more directly to a problem of cognitive control than value-based choice process. The field of cognitive control has studied for many years how distinct demands/context can produce interference in behaviour. Differences that are reflected in the underpinning neural representations. One does not need to go far to find examples, such the extremely well known stroop task, in which irrelevant/incongruent dimension impact behaviour in a way that looks very similar to the effects presented here. In defence, one might argue that what makes this a value-based task is that to different combination of stimuli dimensions was associated different number of points, leading to variable amount of reward, and a feature that the author showed that modulated behaviour. However, this is not dissimilar to manipulating the saliency (or reward) of some perceptual dimensions in a classic cognitive control setting. There are many studies (not discussed here) in which this has been done more or less explicitly

Response:

Thank you for your important feedback.

We fully agree that our task taps into issues of cognitive control and apologize for not having made the relevance of this important topic more clear in our initial version. We now discuss the topic throughout the manuscript (see below for details). While we are happy to highlight cognitive control as an important aspect of our study, we do not believe that this affects the integrity, novelty or importance of our findings. The focus of our study was to go beyond classic distractors and focus specifically on value sensitive neural signals in vmPFC, asking how contextually irrelevant values are organized and represented in this classic value-representing region. We believe that our finding that competing contexts (or goals) and their associated expected values are represented and competing within vmPFC, can help shed light on the importance of values in goal-dependent cognitive control processes (e.g. see Frömer & Shenhav, 2021). One major aspect of our task that is reminiscent of cognitive control is the context switch. We do control for this factor extensively in our results throughout the manuscript. Although our task was not designed to test for the effect of trial since switch, we show that both neural as well as behavioral results survive all corrections for this variable, suggesting that our findings seem to be an inherent feature of value computation rather than a by-product of our specific task setup, as they do reflect more than the strength of required control in a given trial (see our response to Comment 4 below, and e.g., SI Figs. 2, 7, 9 of the manuscript). We are also convinced that our study goes significantly beyond existing studies related to cognitive control. Unlike existing studies, we investigate neural *value* representations associated with outcomes in different contexts. Given that our ROI and classifier were selected and trained in very much the same way many studies investigate value signals in vmPFC, our findings bear significant insights into our understanding of neural value signals. Unlike common cognitive control paradigms, where control helps arbitrate between two features of a chosen object, in our task participants engage control between the values associated with these features in two different task contexts (or states). Moreover our behavioral analyses uncover an effect of the value that the participants could expect if they were in the other context, unlike in stroop or similar paradigms. Computing this value sometimes involves the same choice as the one which is executed, but on other trials it involves the non-chosen option. Hence, the participants must compute, in a counterfactual manner, which choice they would have made, in order to retrieve the irrelevant expected value (termed EVBACK in the paper).

To address the reviewers concern that our findings are not an effect of “manipulating the saliency (or reward) of some perceptual dimensions”, we included an analysis in our manuscript to show that the mere presence of highly rewarding (salient) irrelevant features did not influence participants’ reaction time (no main effect of EVBACK, $\chi^2(1) = 1.21$, $p = .27$, Fig.1a). Rather we find that the value of these features modulated the congruency effect, making incongruent trials slower and congruent trials faster, depending on associated rewards. We also compared the RT model with a EVBACK X Congruency interaction to a model that had a term that reflected the sum of irrelevant values (which could be understood as an overall distraction of the irrelevant context) instead of EVBACK. This analysis showed that the EVBACK model offers a better explanation of the data (AIC EVBACK: -6626.649, AIC Overall Value: -6619.878, Fig.1b). In our analyses of the fMRI data, we also found that the minimum of the two irrelevant features had no additional effect on the strength of value representations (beyond its correlation with EVBACK, LR test with added term $\chi^2(1) = 0.63$, $p = .43$). Finally, we also scrutinized whether perceptual effects alone could explain the effects of EVBACK on classifier probabilities. We reasoned that if the effect of EVBACK on value representation in vmPFC was not because of its associated value, but rather because of reward-induced saliency, then its corresponding perceptual feature would influence vmPFC representation also when it is not the expected value of the ignored context. We therefore tested a model with a parameter that indicates the presence of a perceptual feature corresponding to EVback in the 1D training class (as highest or lowest value; for more details and examples see Fig1d here and Fig.S8 in manuscript). We found no effect of this perceptual parameter (Type II Wald $\chi^2(1) = 0.014$, $p = .905$) indicating that the effect of EVback was indeed due to its associated value, and not value-induced saliency on its corresponding feature. In sum, these analyses show that our

main effects on behavior and fMRI signal in vmPFC are driven by the counterfactual value-dependent EV of the ignored context, and not by the general saliency of the features of the irrelevant context. All discussed effects are illustrated in the figure below.

Figure 1: **a.** Higher expected values of the ignored context (EV_{back}) did not influence participants RT. **b.** The expected value of the ignored context, not the overall value of its features, modulated the Congruency RT effect. **c.** The other value of the ignored context (i.e. the *minimum* of the two features), did not influence EV decodability from vmPFC. **d.** Two example trials to visualize the perceptual similarity control analysis. In both trials the expected value of the cued context is 50 (first column in the table). In the main analysis we modeled the probability associated with the correct class, this class was trained on 1D trials that had the features of 30 and 50 presented in them (second column in table). If the feature corresponding to EV_{back} is capturing the attention and therefore influence representation, then we would expect an increase in decodability when this feature appeared in the corresponding training class. In the first example trial EV_{back} is 30, i.e. appeared in the training class of EV, whereas in the second trial it didn't (right column in table). We found no effect of this perceptual parameter (Type II Wald χ^2 test: $\chi^2(1)=0.014$, $p=.905$)

In the revised manuscript, we now explicitly refer to the cognitive control literature and clarify the relationship between our findings and the stroop effect as well as effects of saliency.

In the introduction (page 2) we write:

“Cognitive control processes are known to arbitrate between relevant and irrelevant information [22, 23], and it has been suggested that they also gate the flow of information within the value network [21, 24]. But although cognitive control does gate relevant information, it is also known that task-switching leads to less than perfect separation between task contexts/goals [23] and results in processing of task-irrelevant aspects [22]. Several studies found traces of the distracting features in several cortical regions, including areas responsible for task execution [25–29]. Similarly, not only task-relevant but also task-irrelevant valuation has been shown to influence cognitive control [30, 31] as well as activity in vmPFC [32] and posterior parietal cortex [33]. We therefore hypothesized that during choice the vmPFC will represent different values that occur in different task contexts, i.e. values appropriate in the current context, as well as other, context-inappropriate and therefore choice-irrelevant values. Importantly, unlike in standard cognitive control settings, we asked whether the above mentioned control during value-based choice involves the arbitration between the expected values that would result from the counterfactual choices one would have made in another context.”

With the additional citations:

21. Amitai Shenhav, Mark A Straccia, Sebastian Musslick, Jonathan D Cohen, and Matthew M Botvinick. Dissociable neural mechanisms track evidence accumulation for selection of attention versus action. *Nature communications*, 9, 2018.
22. Colin M MacLeod. Half a century of research on the stroop effect: an integrative review. *Psychological bulletin*, 109 (2):163, 1991.
23. Stephen Monsell. Task switching. *Trends in cognitive sciences*, 7(3):134–140, 2003.
24. Romy Frömer and Amitai Shenhav. Filling the gaps: Cognitive control as a critical lens for understanding mechanisms of value-based decision-making. *Neuroscience & Biobehavioral Reviews*, 2021..

We also mention the analyses above in the Results:

RT analysis (page 7):

“Neither adding a main effect for EVback nor the interaction of EV × EVback improved model fit (LR-tests: $\chi^2(1) = 1.21$, $p = .27$, $\chi^2(1) = .01$, $p = 0.9$ respectively), indicating that neither the presence of larger irrelevant values alone, nor their similarity to the relevant values influenced participants’ RTs. Additionally, the lower valued irrelevant feature did not show comparable effects and did not interact with Congruency (LR-test to baseline model: $\chi^2(1) = 0.92$, $p = .336$, with interaction: $\chi^2(1) = 2.76$, $p = .251$). Replacing EVback with a parameter of overall value of the irrelevant features did not improve the fit (which could be understood as an overall distraction of the irrelevant context, AIC of model with EVback × Congruency: -6626.649, AIC of model with Overall Value × Congruency: -6619.878, Fig.S3). These results further support that it is specifically the expected reward of the ignored context that played a role in participants’ RT.”

MRI analysis (page 11):

“This analysis revealed that EVback had a negative effect on Pev ... meaning that larger irrelevant expected value led to weaker representation of the relevant one (measured by lower probability of the objective EV, Pev). Importantly, this effect cannot be attributed to attentional effects caused by perceptual input, since replacing EVback with a regressor indicating the presence of its corresponding perceptual feature in the training class, as highest or lowest value, did not provide a better model fit (AICs: -1229.2, -1223.3, respectively, see Fig.S8 for details). Adding the minimum value of the irrelevant context of the trial also did not improve the fit, indicating that it is specifically the highest of the two irrelevant features driving this effect (LR-test with added term: $\chi^2(1) = 0.63$, $p = .43$)...”

And have adjusted our discussion as follows (Pages 17-18):

“One notable aspect of our experiment was that feature relevance was cued on each trial, and rewards were never influenced by irrelevant features. Nevertheless, participants’ behavior was influenced by the expected outcome of the counterfactual choice. This supports the notion that cognitive control based arbitration between relevant and irrelevant features is incomplete [24, 27, 28]. Our neural analyses showed how internal value expectation(s) within vmPFC were shaped by such incomplete suppression: not the ignored context per se influenced vmPFC signals, but rather the computed expected value of the counterfactual choice that would have been made in that context. This was evidenced by the fact that the expected value of the background captured fluctuations in value representations. A control analysis showed that this cannot be explained by the presence of its corresponding perceptual-feature on the screen. Hence, our results cannot be explained by value-independent attention capture caused by the ‘distracting’ irrelevant context (Fig.S8), and go beyond previous research on cognitive control, such as the Stroop Task [22].”

Reviewer 1, comment 4:

Another important feature of this task that further highlights the cognitive control aspect, is that the two contexts alternated very often during the task. This might have exacerbated the interference observed,

making it a by-product of the specific task setup rather than a general feature of value computation. I am indeed curious if the effect of interference was modulated by the lags between the context switch (although given the small range might not be the case). I would like to hear what the authors think and hopefully see some further analyses that might mitigate my worries. In any case, I strongly advise the authors to engage more with cognitive control literature and discuss more in detail these issue and potential caveats, including the problem of rapid switches between contexts. Currently most of the discussion is a recapitulation and summary of the results with very little actual discussion. I believe that the summary can be moved at the end of each result section (improving also clarity – see other point) leaving plenty of space for a more extensive discussion.

Response:

We agree that the effect of the lag between context switches is an important factor to consider in our study.

As mentioned above, we elucidated the behavioral effects of this factor by investigating effects of time since context switch on RT and accuracy in Fig.s S4 and S5 respectively. These analyses revealed a non-significant (but marginal) interaction of switch and congruency ($\chi^2(1) = 3.52, p = 0.068$) on reaction times and a significant effect on choice accuracy, indicating increase in error closer to the switch point ($\chi^2(1) = 13.39, p < 0.001$). Notably, the time since switching did not affect our main behavioral finding (i.e. there was no EVBACK x Congruency X Switch interaction in RT, $\chi^2(1) = 3.70, p = .157$ or accuracy models: $\chi^2(1) = 2.06, p = .356$). Furthermore, our effects also hold when running the models nested within the levels of switch (in a nutshell, this is similar to running each correlation separately within each possible level of switch: once for all the first trials after a context switch, once for the second trials etc. See figure below for RT). Finally, when excluding the trials immediately after a context switch from our analyses (context changed every 4-7 trials, i.e. 32-56 seconds on average) the effects of Congruency and Congruency x EVBACK on RTs still hold (LR-tests: $\chi^2(1) = 8.12, p = .004, \chi^2(1) = 16.61, p < .001$).

We also found no effect of time since switch on the decodability of EV in the vmPFC ($\chi^2(1) = 0.85, p = .36$, Fig.S9). Note, however, that our study does not allow to balance the training set of the classifier with respect to the time since switch. We tried different upsampling methods to decode the switch and could not get any evidence (this also required excluding a fair amount of participants who were missing too many trials for a balanced training set). Since those are null results from an unbalanced design, we decided to not report them in the manuscript.

In summary, aside from time-since-switch effects on error rates, we find no evidence that the rapid alternation between contexts exacerbated the interference effects we observed, particularly not the neural effects in vmPFC that represent the main finding of our paper. Instead, the effects seem to be an inherent feature of value computations.

Nonetheless, we agree that the question of how cognitive control processes relate to context-dependent arbitration during value-based choice is important and discuss this in depth as outlined in response to the previous point. In our discussion, we also mention that follow-up projects could investigate the interplay between task switching and value computations.

We mention these findings in the behavioral results section as follows (page 7):

“All major RT effects hold when running the models nested within levels of EV, Block Context or switch (Fig.S2). Moreover, the number of trials since context switch did not interact with our main effect (LR-test with added term for Congruency×EVback× switch: $\chi^2(1) = 3.70, p = .157$) and our main RT effects still hold when we excluded the first 2 trials after the context switch (LR-tests: Congruency, $\chi^2(1) = 8.12, p = .004$, Congruency×EVback, $\chi^2(1) = 16.61, p < .001$)...”

We added a new panel in Figure S2 for the RT nested models of switch:

Figure 2: Main behavioral RT effect holds nested within the levels of switch. taken from Fig.S2 in the manuscript (new addition). The y-axis represent log(reaction time) of participants and the x-axis expected value of the ignored context (EV_{back}). As can be seen, RT was reduced for congruent trials compared to incongruent, and higher EV_{back} increased this effect, making congruent trials even faster and incongruent trials slower. This effect was also present a few trials into the new context.

And later in the fMRI section (page 12):

“... We also found no effect of time since switch on the decodability of EV (Type II Wald χ^2 test: $\chi^2(1) = 0.85$, $p = .36$, Fig S9, but see discussion on limitations)...”

And address this in the discussion (page 18):

“Participants repeatedly had to switch between contexts in our task, a process that is well known to engage cognitive control mechanisms [21–24, 31]. We evaluated to what extent this task switching affected our results and found that behavioral effects hold when excluding the first 2 trials after a context switch, and that the distance from the last switch did not interact with the influence of the irrelevant values (Fig.S2). Likewise, we found no influence of task switching on multivariate EV effects in vmPFC. Note, however, that due to our design we could not create balanced training sets (with respect to number of trials since context switch) which would be required for a more thorough investigation of the effect of trials since switch on value signals. We therefore conclude that while context switching is part of the investigated phenomenon, its presence alone cannot explain our findings.”

Reviewer 1, comment 5:

My main worry is that without more empirical evidence and extensive discussion about this issue the interpretation of these results might result misleading for the field of value-based decisions (I am sure that this is not the author intended).

I hope that the authors will find my feedbacks useful.

Response:

Thank you for your feedback. We found it very useful to clarify and streamline the manuscript and hope we provided a comprehensive answer to your concerns.

Reviewer 3

Reviewer 3, comment 1:

Moneta et al. examine how context and value are represented in vmPFC during a contextual value-based decision making task. They show that multivariate patterns in vmPFC contain information about context and contextual value. Furthermore, they leverage behavioral impacts of context-irrelevant values and trial-by-trial analyses of decoder accuracy to examine how context-irrelevant value information can interfere with stored values when context representations are weak.

The paper is interesting, timely, and the research is generally well designed and described. It uses neuroimaging analyses innovatively to tackle mechanistic questions that are typically off limits for human studies, and I think that it makes great strides in doing so. That said, I have a number of concerns with the paper in its current form, as well as questions about what exactly was done, and why. A complete listing of my comments is below.

Response:

Thank you for your positive feedback.

Reviewer 3, Major comment 1:

I was a bit surprised to see that the authors do not perform deconvolution to attain single trial activations for the multivariate analyses, but instead use a single TR at the peak of the HRF. This approach seems like it could have at least two potential shortcomings. **One** is that timing variability could be mis-attributed to amplitude variability. It appears the authors do not have jitter in their task timings (though it would be useful to state this explicitly). Even so, the participant responses are variable, making it impossible to get around this issue completely. The **second** shortcoming is that it leaves potentially informative data on the table – that is to say that TRs before and after the peak may provide some additional information about the activation that is not picked up using the current methods. The degree to which this is true is complicated, as it depends on the temporal autocorrelation of both signal and noise, as well as task design, but at minimum it would be useful for the authors to provide some justification for their choice, and it would be even more compelling if they could demonstrate that the key effects hold when using the deconvolution approach.

Response:

Thank you for this important comment. Firstly we apologize that it was not made clear in the manuscript that events indeed haven't been jittered in our task. The time between cue and stimulus was drawn from a truncated exponential distribution with a mean of 0.6s (range: 0.5s-2.5s). The same was true for the time between stimulus and outcome with a mean of 3.4s (range 1.5s-9s) and between outcome and next cue with a mean of 1.25s (range 0.7s-6s). The information was only present in the online methods and not in Figure 1. We now added this important information to Figure 1 and the caption.

We acknowledge that deconvolution has notable benefits (Hinrichs et al, 2000), but we note that the major benefits of deconvolution are seen when dealing with severely overlapping or sequential events. In order to test that the jittering between events in our task was sufficient to prevent such overlap, we verified the estimability of the design matrices by assessing the Variance Inflation Factor (VIF) for each subject. This showed that even when setting up a design matrix with one regressor per trial, none of the VIFs surpassed a value of 3.5 (a value of 5 is considered a conservative indicator for overly collinear regressors, see for example Mumford et al., 2015). Hence, taking a single datapoint at a fixed time after stimulus onset is very unlikely to have resulted in activity estimates conflated by other task events (see “Verifying design trial-wise estimability” page 31 in online methods). In addition, we want to note that using TRs rather than

deconvoluted patterns or GLM estimated betas is very common practice in fMRI decoding (e.g. to name a few papers: Polyn et al., 2005, *Science*; Momennejad et al., 2017, *eLIFE*, Wittkuhn & Schuck 2021, *Nature Communications*). Hence, we do not see any major concern with the approach we have chosen.

Nevertheless, we want to address the reviewer's concerns about activations before and after the peak, as well as the influence of RT-related variability. We therefore decoded values using trial-wise betas (one regressor per trial with duration set to RT, set up using an approach that involves one GLM per trial and subject, as suggested by Mumford et al., 2015, *Plos One*). This approach allowed us to decode EV and Context from the vmPFC betas above chance (t test against chance for EV: $t(34) = 4.32$, $p < 0.001$ and for Context: $t(34) = 5.25$, $p < 0.001$), also when tested only on 2D trials, as in the manuscript (EV: $t(34) = 3.8$, $p < 0.001$ and for Context: $t(34) = 4.68$, $p < 0.001$). But we found no effects of EVBACK or Congruency (Type II Wald χ^2 test: EVBACK $\chi^2(1) = -0.04$, $p = 0.83$; ; Congruency $\chi^2(1) = 0.19$, $p = 0.655$). Notably, we also found a very strong effect of $\log(\text{RT})$ on the decodability of the classifier (Type II wald χ^2 test: $\chi^2 = 32.96$ $p < 0.001$). Running a similar trialwise GLM analysis with event duration set to 0 allowed us to decode EV and Context (t test against chance for EV: $t(34) = 4.4$, $p < 0.001$ and for Context: $t(34) = 4.47$, $p < 0.001$), but also showed less sensitivity to our main effects of EVBACK and Congruency (Type II Wald χ^2 test: EVBACK $\chi^2(1) = 0.28$, $p = 0.59$; Congruency $\chi^2(1) = 0.11$, $p = 0.73$).

Compared to these results, the approach we have chosen in our paper does not suffer from any of these drawbacks. First, the classifier trained on raw data was not affected by RT variability in the way the GLM approach was (Type II wald χ^2 test: $\chi^2 = 0.0540$ $p = 0.8162$). Hence, directly incorporating RT variability into a GLM approach seems to introduce a RT confound, rather than solving it, while our approach of taking the raw data 5 seconds after onset in combination with jittered trial timing helped to minimize RT effects. This approach of not taking RT effects into account is in line with previous research on value decoding e.g. in McNamee, Rangel & O'Doherty, 2013 *Nature Neuroscience*. In addition, our classifier seems to have greater sensitivity to the effects of EVBACK and linked to behavioral Congruency effects, which are the main interest of our paper. Most importantly, we would also like to note that our main finding that EVBACK decreases EV decodability does not appear in participants' RT (no main effect of EVBACK on RT). At the same time, our main RT effects do not appear in the fMRI decodability (no Congruency effect on EV).

To confirm that the reported effects of EVBACK and Congruency on value activations can be recovered using an independent method, we used GLM-derived betas in an RSA analysis, suggested by the reviewer below (comment 2). In this analysis, we fitted a GLM with one separate regressor for each combination of EV and EVBACK, irrespective of the context (cross-validated, 1D trials modeled separately, see below for details). In this analysis, we find a clear main effect of EVBACK representation in vmPFC and also confirm our main finding that higher EVBACK reduces the similarity of EV representation. These novel results are now reported in the manuscript (see response to reviewer 3, comment 2 for details). In conclusion, we are therefore confident that our methods are appropriate, robust and sensitive to the phenomenon of interest, either when looking at raw data or a deconvolution approach. We have added a statement to the discussion in which we mention the possible usefulness of deconvolution approaches for future research (page 19):

"Another methodological aspect was that we decoded based on timeshifted TR images, rather than deconvolved activity patterns [51] as is common practice in fMRI decoding papers [17, 52–54]. Decoding level and approach may have implications for the representations that can be uncovered in future research."

Reviewer 3, Major comment 2:

Some key reported results rely on the degree to which patterns of activation in vmPFC resemble those most diagnostic of high EV_back trials. Yet, there is very little evidence to suggest that vmPFC represents EV_back. Decoding of EV_back category was not significantly above chance, and thus interpreting the key results that rely on EV_back classifiers seems a bit tricky. Is EV_back represented in vmPFC, as the authors

suggest... or does training a classifier on EV_back for some reason pick up on noise dimensions that have some interesting relationship to EV, and therefore behavior? The original design was balanced, but the need to oversample specific trials makes this a distinct possibility, and thus a bit hard to tell. Could the authors demonstrate successful classification of EV_back in specific individuals? Or perhaps do analyses that would allow them to understand why EV_back might not be decodable, even if the information were present? I could imagine other methods for detecting multivariate representations, such as RSA, could be helpful in this regard. As it stands, I think that there is a lot of ambiguity on how to interpret results that rely on the EV_back classifier, and if additional support for the representation of EV_back in vmPFC is not provided, then specific mechanistic claims would need to be substantially scaled back.

Response:

We would like to thank the reviewer very much for this comment.

First, we would like to note that we show two independent effects that indicate a representation of EVBACK. In one analysis (Fig5a-f), we show how the probability of the value classifier is affected by EVBACK. Note that this classifier is fully balanced w.r.t. the frequency of the main classes, and that it was trained on 1D trials, in which no background values were present. Hence it could not have picked up on noise related to EVBACK, and balancing issues could not have affected our results. In a second, independent, analysis (Fig5g-i), we trained a classifier directly on the EVBACK of 2D trials (i.e. using different data than the previous analysis) and investigated the probability this classifier assigned to each EVBACK class...

The reviewer is correct that the balancing of the second analysis represents a major obstacle. As can be seen in the figure below, we had many more unique trials in which EVBACK was 70 than 50 or 30. A close inspection of the classifier predictions showed that despite our up-sampling attempts, the EVBACK classifier still showed a corresponding frequency bias (lower average probability for the less frequent classes). To account for this effect, we modeled the effects of variables of interest on the EVBACK probabilities nested within the EVBACK levels, meaning that the mean difference due to the frequency of the classes cannot account for the fixed effects of the model (in a nutshell, this is similar to running each correlation separately within each level of EVBACK). In this analysis, we confirmed the previous finding that an increase in neural representation of EVback reduced EV decodability (lowered AIC score from -1223.6 to -1225.0, but note that in the LR-test $\chi^2(1) = 3.45$, $p = 0.063$). Most remarkably, we show that the stronger the relationship between Context and EV representations, the less vmPFCs irrelevant value signal competed with its value representations, akin to a shielding effect (LR-test with $P_{evback} \times P_{context}$ interaction effect: $\chi^2(1) = 5.22$, $p = 0.022$).

Figure 3: **a.** EVBACK frequency bias within the design. For every pair of features in the relevant context, we varied the irrelevant features. As can be seen, our design had more trials for higher EVBACK. Taken from Figure 1 in the manuscript. **b.** A classifier trained directly on EVBACK of 2D trials was biased by the frequency described in panel a and assigned lower probability to the less frequent classes. This has been overcome using mixed effect models nested in the levels of EVBACK as well as in the RSA analyses

Finally, we also want to address reviewers' concerns about the lack of direct evidence. We therefore ran RSA analyses in a way that allowed for the investigation of EVBACK effects. The RSA used betas estimated from HRF convolved regressors in a GLM (major comment1). In short, we fitted a GLM with one separate regressor for each combination of EV and EVBACK, irrespective of the context (cross-validated, 1D trials modeled separately). After multivariate noise normalization and mean pattern subtraction (Walther et al. 2016), we computed the Mahalanobis distance of each regressor combination resulting in a 9 by 9 Representational Dissimilarity Matrix (RDM, see Figure below and manuscript). We then fitted mixed effects models to test whether pattern dissimilarities are sensitive to EV and EVBACK. Indeed, as expected, adding a main effect for EV dissimilarity (0 when two regressors share the same EV, 1 otherwise) improved model fit compared to a null model (LR-test: $\chi^2(1)=10.89$, $p<.001$, panel a). Strikingly, adding a main effect of EVBACK dissimilarity (0 when sharing EVBACK and 1 otherwise) further improved model fit (LR-test with added term: $\chi^2(1)=247.67$, $p<.001$, panel b). We believe this provides direct evidence of EVBACK representations in vmPFC and would like to thank the reviewer for making this helpful suggestion.

Crucially, because of the symmetric nature of the RSA, in this analyses the effect of the frequency of EVBACK classes (which biased the classifier) is orthogonal to the main effect of EVBACK and its value similarity - as can be seen in panels c and d.

In the manuscript, we also replicated the value-similarity effect where we show that similar values also show more similar patterns (e.g. 30 and 50 are more similar than 30 and 70). We now show the same effect also with respect to EVBACK (which is also orthogonal to the frequency bias).

Figure 4:

RDM analyses showing clear evidence for EV and EVBACK representation within the vmPFC (taken from Fig.3 and S6 in the manuscript). a. Betas that shared the same EV were less dissimilar than betas that didn't. b. Betas that shared the same EVBACK were less dissimilar than betas that didn't. c. Frequency bias within the RDM: Each cell shows the number of how many trials were used for to both the betas that correspond to that cell (presented as ratio relative to the rest). d. Correlation of all parameters used in the RDM analysis shows that our main effect of EVBACK similarity is orthogonal to the frequency.

Lastly, in the RDM models we also replicate the main effect of EVBACK on value signals, showing that higher EVBACK reduces the EV similarity. Specifically, when only modeling trials that share the same EV (where dissimilarity should be at its lowest, dark cells in panel a), when EVBACK was high, the dissimilarity increased (Type II Wald χ^2 test: $\chi^2(1)=36.6$, $p<.001$, also in Fig. S6).

We are excited about these new results and have therefore decided to include them in the main manuscript, page 8,9 and 10, as well as in the methods on pages 31-32. The key passage and the new figure are found below:

Outcome-relevant and outcome-irrelevant values co-exist within the vmPFC

We derived a value-sensitive vmPFC ROI following common procedures in the literature [e.g. 4, 5] (see Fig.3a, and Methods) and tested whether both relevant and irrelevant expected values are reflected in multivariate vmPFC patterns using a RSA framework. To estimate value-related activity patterns within the vmPFC mask, we fitted a General Linear Model (GLM) with one separate regressor for each combination of EV and EVback, irrespective of the context (cross-validated, 1D trials modeled separately). After multivariate noise normalization and mean pattern subtraction ([see 48]) we computed the Mahalanobis distance between each combination of regressor. This resulted in one 9×9 Representational Dissimilarity Matrix (RDM, Fig.3 and online methods) per subject, which we analyzed using mixed effects models (Gamma family with a inverse link, [49]). We first asked whether EV was reflected in the RDMs, as expected given that we used a functionally defined value ROI. Indeed, adding a main effect for EV dissimilarity (0 when two

regressors share the same EV, 1 otherwise) improved model fit compared to a null model (LR-test: $\chi^2(1) = 10.89$, $p < .001$, Fig.3b). Next, we asked if the activity patterns from trials with the same EVback were more similar than than patterns reflecting different EVback. Strikingly, adding a main effect of EVback dissimilarity (0 when sharing EVback and 1 otherwise) further improved model fit (LR-test with added term: $\chi^2(1) = 247.67$, $p < .001$, Fig.3c.).

We then reasoned that the neural codes of expected values should also reflect value-differences in a gradual manner. We therefore asked whether pattern similarity was not only increased if two trials had the same value (e.g. comparing '30' to '30', Fig.3d. purple cells), but also higher when the values in two trials had a difference of 20 (e.g. '30' to '50', Fig.3d. turquoise) compared to a value difference of 40 (e.g. '30' to '70', Fig.3d. yellow). Indeed we found that adding main effects for the value difference of EV as well as EVback improved model fit (VDEV :LR-test compared to a null model: $\chi^2(1) = 12.34$, $p < .001$, VDEVback : LR-test with added term: $\chi^2(1) = 256.98$, $p < .001$, Fig.3c-d.). Note that the full model with both value difference effects resulted in a better (lower) AIC score than the model with both main effects of the EVs (AIC=165231 and AIC=165241, respectively, 209 Fig.S6) indicating that the value similarity effect is not merely driven by the diagonal. Full models including effect sizes and confidence intervals can be found in SI TableS5 and TableS6.

Hence, neural patterns in vmPFC were affected 212 by contextually-relevant as well as irrelevant value expectations. Notably, the values of irrelevant features were computed despite being counterfactual (not related to the choice), and co-existed with well known expected values signals in vmPFC.

Reviewer 3, Major comment 3:

I was a bit confused as to why the authors use categorical classification procedures throughout. The authors seem to think the vmPFC is representing a continuous value signal – so why not predict continuous value from multivariate activation, rather than the category? Doing so would directly assess the dimensions over which appropriate context and inappropriate context values are represented (ie. the weights from the predictive model). In this case the authors could just examine the cosine angle between these dimensions to see whether representation of the two value signals (context appropriate/context inappropriate) is orthogonal in those participants who show less interference, and perhaps non-orthogonal in those participants who show more.

Response:

One of the goals of the project was to investigate the interaction between a context (or state) signal and value signals. We believe that vmPFC represents a state signal (e.g., Schuck et al., Neuron, 2016), whereby a continuous value representation might be part of the signal, but not the only part. Because aspects of the state (the context) can not be represented continuously, we decided to choose a categorical classifier that could be used for the value and the task context analysis avoiding unnecessary differences between the two decoders. We now refer to this directly in the discussion. Nevertheless, we believe that the value similarity analysis (both in the RDM and classifier probabilities) additionally shows evidence for such continuous value representation, yet still captures other task aspects that might not be continuously represented.

We now refer to this reasoning in the discussion (page 19):

“Another important implication of our study concerns the nature of neural representations in vmPFC/mOFC, and in particular the relationship between state [34, 36, 37, 50] and value [2–7] codes in this area. In order to compare both aspects, we used a categorical classifier for value as well as states, rather than examining continuous value representations. Nevertheless, we believe that the value similarity analysis (both in the

RSA, Fig.3d-e. and classifier probabilities, Fig.S7) additionally shows evidence for such continuous value representations. We specifically chose to focus on the vmPFC region that is commonly investigated in value-based decision research. We therefore defined our ROI in a univariate manner as commonly done in the literature [e.g. 4, 5]) and studied the multivariate state and value signal within this ROI [e.g. 34, 36] ...”

Reviewer 3, Major comment 4:

Even if the authors do have a good reason not to use a continuous decoding strategy – it still seems like it would be useful to understand the relationship between the multivariate codes that they measure. For example, in existing models, do the readout weights for the highest EV look similar to those for the highest EV_back? Or are they completely independent?

Response:

The reviewer raises an interesting point about the nature of the patterns that underlie our effect. As previously mentioned, both RSA and classification analyses show evidence for continuous value representations in the vmPFC. Moreover, our value classifier was sensitive to EV of the irrelevant context, even though it was trained on 1D trials, i.e. trials where there were no irrelevant values. This suggests some similarity between the encoding of relevant and irrelevant values within the vmPFC. However, we note that the interpretation of weight vectors from decoding approaches is not straightforward, and prevents us from making the inferences the reviewer asks about (see Haufe et al., NeuroImage, 2014). We therefore would like to refrain from inspecting and interpreting the weight vectors.

We refer to this interpretation in the discussion (page 18):

“... our analysis showed that the value classifier was sensitive to the expected value of the irrelevant context in 2D trials, even though it was trained on 1D trials during which irrelevant values were not present. This suggests that within vmPFC ‘conventional’ expected values and counterfactual values are encoded using partially, but not completely, similar patterns.”

Reviewer 3, Major comment 5:

Throughout manuscript, there were p values reported without adequate description of the statistical test and null hypothesis. For example, in figure 1a, but also throughout main text. In some cases, statistical tests were implied (Chi squared values reported, and presumably likelihood ratio tests were performed) but I was a bit unsure why – since in some cases both variables examined were continuous and models nested, such that I expected to see a nested-F test. Clarification and adequate explanation in main text would be helpful.

Response:

Thank you for pointing out this important aspect. We now mention next to every p-value which test it refers to. Due to word limitation of figure caption, within it we only mention the p values, but we made sure that every test mentioned in the figures is described in detail in the main text.

Additionally, in the online methods (page 33) we clearly state which statistical test we used :

“When describing main effects of models, the χ^2 represents Type II Wald χ^2 tests, whereas when describing model comparison, the χ^2 represents the log-likelihood ratio test. Model comparison throughout the paper was done using the ‘anova’ function...”

Regarding the F test for nested models, the F test assumes the data is normally distributed. And whereas that is the assumption in the RT models, we can not make this assumption when modeling the probabilities of the classifier or the RSA similarity measures (as they are beta and gamma distributed). This is why we used gamma and beta mixed effects models, as appropriate, using the toolbox glmmTMB.

The reasoning behind the choice to perform Chi Square test is now explained in the methods section as follows (page 26):

“...*The reason we used χ^2 test is that classification probabilities as well as RSA dissimilarities are not normally distributed (these follow beta and gamma distributions respectively, note that the glmmTMB toolbox also uses χ^2 as its default for these distributions).*”

Reviewer 3, Major comment 6:

In general, AIC values without any associated uncertainty measures or confusion matrices to validate them are a bit tricky to interpret. I also found likelihood ratio test statistics a bit tough to interpret – because in many cases the direction of the coefficients assigned to the coefficient of interest was more critical to the overarching hypothesis than the amount of variance explained. In most cases I think that relevant parameter estimates, along with their 95% confidence intervals, would be a more useful way to display results from the statistical modeling.

Response:

We agree that it is important to provide directional information about the reported effect. We made sure that the directions of all effects are stated in the text and/or are displayed in the figures. As our paper is already very dense, we added SI tables with all parameter estimates and 95% confidence intervals of all the main effects reported, one for each best explaining model (RT, the two RSA analyses, and models on EV decodability). The table can be found below and on pages 70-71 in the manuscript. We also made sure that the direction of the effects is evident from plots throughout the paper, and/or mentioned in the text:

Effect sizes and confidence intervals for best explaining models In all the following tables, CI marks Confidence Interval, CIl and CIh the low and high ends of the confidence interval respectively.

Table S2: Effect sizes and confidence intervals for best explaining RT model

Parameter	Coef	CI	CIl	CIh	t	df	er	p	Effects	Std _{Coef}	Std _{Coef} ^{CIl}	Std _{Coef} ^{CIh}
1 (Intercept)	0.08	0.95	0.06	0.11	6.48	9035	0.00	fixed		0.17	0.04	0.29
2 Switch	-0.01	0.95	-0.01	-0.01	-6.57	9035	0.00	fixed		-0.06	-0.08	-0.04
3 Trial	-0.01	0.95	-0.02	-0.01	-6.92	9035	0.00	fixed		-0.06	-0.08	-0.04
4 side [R]	-0.01	0.95	-0.01	-0.00	-2.29	9035	0.02	fixed		-0.04	-0.08	-0.01
5 Context [M]	-0.07	0.95	-0.08	-0.07	-20.73	9035	0.00	fixed		-0.37	-0.40	-0.33
6 EV	-0.07	0.95	-0.08	-0.07	-29.14	9035	0.00	fixed		-0.36	-0.39	-0.34
7 Congruency [1]	0.02	0.95	0.01	0.03	5.40	9035	0.00	fixed		0.10	0.06	0.13
8 Congruency [-1] * EV _{back}	-0.01	0.95	-0.01	-0.00	-2.07	9035	0.04	fixed		-0.03	-0.05	-0.00
9 Congruency [1] * EV _{back}	0.01	0.95	0.00	0.01	3.73	9035	0.00	fixed		0.05	0.02	0.07
10 EV * Congruency [1]	0.01	0.95	0.00	0.01	2.08	9035	0.04	fixed		0.04	0.00	0.07
11	0.07	0.95						random-sub				
12	0.17	0.95						random-Residual				

Table S3: Effect sizes and confidence intervals for best explaining fMRI model (main model)I

Parameter	Coef	CI	CIl	CIh	z	p	Effects	Std _{Coef}	Std _{Coef} ^{CIl}	Std _{Coef} ^{CIh}
1 (Intercept)	-0.65	0.95	-0.69	-0.61	-30.83	0.00	fixed	-0.65	-0.69	-0.61
2 EV _{back}	-0.05	0.95	-0.08	-0.01	-2.58	0.01	fixed	-0.05	-0.08	-0.01
3 logit(P _{context})	0.06	0.95	0.02	0.10	3.25	0.00	fixed	0.06	0.02	0.10
4	0.06	0.95	0.02	0.18			random-sub			
5	8.79	0.95					random-Residual			

Table S4: Effect sizes and confidence intervals for best explaining fMRI model (model nested in EV_{back})

Parameter	Coef	CI	CII	CIh	z	p	Effects	Std _{Coef}	Std ^{CI} _{Coef}	Std ^{CIh} _{Coef}
1 (Intercept)	-0.65	0.95	-0.69	-0.61	-31.36	0.00	fixed	-0.65	-0.69	-0.61
2 logit(P _{context})	0.06	0.95	0.02	0.09	3.02	0.00	fixed	0.06	0.02	0.09
3 mlogit(P ^{EV_{back}})	-0.03	0.95	-0.07	0.01	-1.63	0.10	fixed	-0.03	-0.07	0.01
4 logit(P _{context}) * mlogit(P ^{EV_{back}})	0.04	0.95	0.00	0.08	2.17	0.03	fixed	0.04	0.00	0.08
5	0.05	0.95	0.00	0.62			random-EV _{back} :sub			
6	0.04	0.95	0.00	0.70			random-sub			
7	8.82	0.95	8.16	9.54			random-Residual			

Table S5: Effect sizes and confidence intervals for best explaining RSA model - diagonal models

Parameter	Coef	CI	CII	CIh	z	df_error	p	Effects	Std _{Coef}	Std ^{CI} _{Coef}	Std ^{CIh} _{Coef}
1 (Intercept)	0.01	0.95	0.01	0.02	30.66	Inf	0.00	fixed	0.01	0.01	0.02
2 EV ^{diagonal}	-0.00	0.95	-0.00	0.00	-1.77	Inf	0.08	fixed	-0.00	-0.00	0.00
3 EV ^{diagonal} _{back}	-0.00	0.95	-0.00	-0.00	-15.61	Inf	0.00	fixed	-0.00	-0.00	-0.00
4	0.00	0.95	0.00	0.00				random-freq:sub			
5	0.00	0.95	0.00	0.00				random-sub			
6	0.27	0.95	0.27	0.27				random-freq			

Table S6: Effect sizes and confidence intervals for best explaining RSA model - value difference models

Parameter	Coef	CI	CII	CIh	z	df_error	p	Effects	Std _{Coef}	Std ^{CI} _{Coef}	Std ^{CIh} _{Coef}
1 (Intercept)	0.01	0.95	0.01	0.02	30.66	Inf	0.00	fixed	0.01	0.01	0.02
2 VDev	-0.00	0.95	-0.00	-0.00	-2.17	Inf	0.03	fixed	-0.00	-0.00	-0.00
3 VDev _{back}	-0.00	0.95	-0.00	-0.00	-15.93	Inf	0.00	fixed	-0.00	-0.00	-0.00
4	0.00	0.95	0.00	0.00				random-freq:sub			
5	0.00	0.95	0.00	0.00				random-sub			
6	0.27	0.95	0.27	0.27				random-freq			

Reviewer 3, Minor comment 1:

I was a bit confused by the logic relating to the predicted interaction between context, EV_{back}, and EV (lines 348-9). Is the idea that you expect to see EV_{back} representations to have minimal contribution to the accuracy of EV representations when context representations are confident and accurate, but more influence when they are inaccurate? This was originally my interpretation, but this is not clear from the lines referenced above, and I think a schematic illustration of the author’s prediction along with a display of the results in the same format could be helpful here.

Response:

Yes. The reviewer understood correctly. Following the reviewer’s suggestion, we adapted the corresponding paragraph and added an illustration of the relationship between the two EVs in Figure 5 which we hope will help clarify our predictions.

The paragraph the reviewer refers to was updated as follows (page 13):

“Most remarkably, the effect of Context, Pcontext, interacted with the effect of P2D_EVback, such that when the context signal was stronger, the negative effect of irrelevant value signals on relevant value signals was weaker (i.e. Pcontext affected the association between P^{(2D)_evback} and P_{ev}, LR-test: $\chi^2(1) = 5.22$, $p = 0.022$, Fig.5e). In other words, the stronger the relationship between Context and EV representations, the less vmPFCs irrelevant value signal competed with its value representations, akin to a shielding effect”

And later we clarify further (page 13):

“In summary... Most strikingly, the negative influence of EV_{back} representation on EV decodability was mediated by a neural context signal, i.e. when the link between Context and EV increased, the effect of EV_{back} representations diminished.”

The new illustration and new text from Fig. 5 where we clarify the predicted negative relationship between the two classes corresponding to EV and EVBACK. We believe that understanding this relationship helps understand the following interaction presented later in the figure:.

Figure 5, panels (a) and (d): Left: Illustration of a 2D color trial (top), the outcomes associated with the relevant colors (middle), and the outcomes associated with the irrelevant motion features (bottom). The maximum of relevant color outcomes is termed EV; the maximum of the irrelevant motion outcomes is termed EVback. Right: Illustration of value classifier probabilities, reflecting EV (P_{EV}), EVback (P_{EVback}) and the outcome that is neither EV or EVback (P_{Other}) in example trial shown in the left panel.

Reviewer 3, Minor comment 2:

In Figure 1e it would be useful to clarify in figure that the matrix displayed is representing the context-irrelevant feature pairs. This info is in caption, but could be a bit more clearly communicated.

Response:

We added a clarification to the figure.

Reviewer 3, Minor comment 3:

Throughout paper, there are plots that include $\text{logit}(\text{prob})$ on the x-axis in which model predictions are displayed, however it is unclear the range of that axis that actually contains data. It would be useful if some markers for the range of the actual data could be provided, so the magnitude of the effects easily interpreted.

Response:

Thank you for pointing this out. We now added individual data (showing the range of the actual data) for figures 4f (positive relationship between EV and Context decodability) and 5g-h (negative relationship between EV and EVBack decodability, modulated by context decodability). We note that for the interaction panel of Fig 5h and for Fig 6, the visualization of the data is complicated in a single panel which is why we only show the range there (as requested by the reviewer) and added two additional SI figures (S10,S11) for the individual lines.

The Figures S10 and S11 are shown below:

Figure 6: addition of data to panels from manuscript in order to clarify the range of predictors. Panel a is taken from figure 4f, panel b and c (left) from figure 5g-h in the manuscript. Panel c appears as figure S10 in manuscript. Full caption see manuscript.

Figure 7: addition of data to panels from manuscript in order to clarify the range of predictors. Note that to the request of the reviewer and for illustration purposes, only the range of each predictor (x axis) was added to figure 6. The left side of each panel is taken from figure 6, whereas the full figure as presented here can be found as S11 in the manuscript (Full caption see manuscript).

Reviewer 3, Minor comment 4:

I think that there are a couple of references to figure 5 that should actually be figure 4.

Response:

Thank you. Now all the references point to the corresponding figure.

Reviewer 3, Minor comment 5:

Line 345: independent is probably not the right word here.

Response:

As previously mentioned (major comment 2), since the predictors in those analyses come from two separate classifiers that were trained on different trials (1D vs 2D trials) we believe the word independent is the correct one. We hope the above response and the formulation in the new manuscript help clarify this issue.

Reviewer 3, Minor comment 6:

The terminology used to describe trial-to-trial variability in classifier reports could use improvement. Terms like “strongly decoded” seem misleading, given overall decoding accuracy, and that what they are really trying to say is that this trial has a high amount of some multivariate feature.

Response:

Thank you for this important comment. When describing effects of the multivariate analyses we changed the terminology (mostly to “increased probability” or “increase in strength of representation” or “more evidence for..”).

Reviewer 3, Minor comment 7:

In figure 6a, decoding of pEV_back for congruent trials is at (or below) chance – but presumably this is only because the classifier was trained on all trial types, right?

Response:

Yes the reviewer is correct. This is because those are trials where participants made a ‘wrong’ choice, i.e. did not choose the cloud with the maximum EV or the maximum EVback.

Reviewer 3, Minor comment 8:

Figure 6b: legend would be helpful in this figure.

Response:

We believe the reviewer meant Figure 6c (because panel b had a legend already). We added a legend to the panel.

Reviewer 3, Minor comment 9:

The authors summary states that accuracy and RT data support their claims – but I do not see results from choice regression in main paper, unless I have missed something.

Response:

The results of the choice regression appeared at the end of the behavioral results section and in Fig. S5. Indeed, we did not refer to the SI figure in the main text before and we have now corrected that (Page 8):

“...We next modeled choice accuracy in 2D trials using the same analysis approach and nuisance variables (see methods and Fig.S5) and found the same effects as the RT models: (1) Higher accuracy for higher EV (LR-test: $\chi^2(1) = 14.61, p < .001$) (2) decreased performance on incongruent trials with (3) higher error rates occurring on trials with higher EVback (LR-tests: $\chi^2(1) = 66.12, p < .001, \chi^2(1) = 6.99, p = .03$, respectively, Fig.S5).”

References:

Frömer, R., & Shenhav, A. (2021). Filling the gaps: Cognitive control as a critical lens for understanding mechanisms of value-based decision-making. *Neuroscience & Biobehavioral Reviews*.

Hinrichs, H., Scholz, M., Tempelmann, C., Woldorff, M. G., Dale, A. M., & Heinze, H. J. (2000). Deconvolution of event-related fMRI responses in fast-rate experimental designs: tracking amplitude variations. *Journal of cognitive neuroscience*, 12(Supplement 2), 76-89.

Jeanette A Mumford, Jean-Baptiste Poline, and Russell A Poldrack. Orthogonalization of regressors in fmri models. *PloS one*, 10(4):e0126255, 2015.

Polyn, S. M., Natu, V. S., Cohen, J. D., & Norman, K. A. (2005). Category-specific cortical activity precedes retrieval during memory search. *Science*, 310(5756), 1963-1966.

Momennejad, I., Otto, A. R., Daw, N. D., & Norman, K. A. (2018). Offline replay supports planning in human reinforcement learning. *elife*, 7, e32548.

Wittkuhn, L., & Schuck, N. W. (2021). Dynamics of fMRI patterns reflect sub-second activation sequences and reveal replay in human visual cortex. *Nature communications*, 12(1), 1-22.

McNamee, D., Rangel, A., & O'doherty, J. P. (2013). Category-dependent and category-independent goal-value codes in human ventromedial prefrontal cortex. *Nature neuroscience*, 16(4), 479-485.

Walther, A., Nili, H., Ejaz, N., Alink, A., Kriegeskorte, N., & Diedrichsen, J. (2016). Reliability of dissimilarity measures for multi-voxel pattern analysis. *Neuroimage*, 137, 188-200.

Haufe, S., Meinecke, F., Görgen, K., Dähne, S., Haynes, J. D., Blankertz, B., & Bießmann, F. (2014). On the interpretation of weight vectors of linear models in multivariate neuroimaging. *Neuroimage*, 87, 96-110.

Schuck, N. W., Cai, M. B., Wilson, R. C., & Niv, Y. (2016). Human orbitofrontal cortex represents a cognitive map of state space. *Neuron*, 91(6), 1402-1412.

REVIEWERS' COMMENTS

Reviewer #1 (Remarks to the Author):

I am happy with the replies to my comments. They did a careful and thoughtful job, well done. This is a very good paper and in my opinion ready for publication.

Reviewer #3 (Remarks to the Author):

The authors have fully addressed my concerns.